# The Impact of Footwear on Occupational Task Performance and Musculoskeletal Injury Risk: A Scoping Review to Inform Tactical Footwear

**DOI:** 10.3390/ijerph191710703

**Published:** 2022-08-27

**Authors:** Robin Orr, Danny Maupin, Robert Palmer, Elisa F. D. Canetti, Vini Simas, Ben Schram

**Affiliations:** Tactical Research Unit, Faculty of Health Sciences and Medicine, Bond University, Gold Coast, QLD 4226, Australia

**Keywords:** work boots, work shoes, worker safety, worker health, injury prevention

## Abstract

The aim of this scoping review was to investigate the impact of footwear on worker physical task performance and injury risk. The review was guided by the Preferred Reporting Items for Systematic Reviews and Meta-Analyses extension for Scoping Reviews protocol and registered in the Open Science Framework. Key search terms were entered into five academic databases. Following a dedicated screening process and critical appraisal, data from the final articles informing this review were extracted, tabulated, and synthesised. Of 19,614 identified articles, 50 articles informed this review. Representing 16 countries, the most common populations investigated were military and firefighter populations, but a wide range of general occupations (e.g., shipping, mining, hairdressing, and healthcare workers) were represented. Footwear types included work safety boots/shoes (e.g., industrial, gumboots, steel capped, etc.), military and firefighter boots, sports shoes (trainers, tennis, basketball, etc.) and various other types (e.g., sandals, etc.). Occupational footwear was found to impact gait and angular velocities, joint ranges of motion, posture and balance, physiological measures (like aerobic capacity, heart rates, temperatures, etc.), muscle activity, and selected occupational tasks. Occupational footwear associated with injuries included boots, conventional running shoes, shoes with inserts, harder/stiffer outsoles or thin soles, and shoes with low comfort scores—although the findings were mixed. Occupational footwear was also linked to potentially causing injuries directly (e.g., musculoskeletal injuries) as well as leading to mechanisms associated with causing injuries (like tripping and slipping).

## 1. Introduction

Certain occupations exist that work in potentially dangerous environments, such as, but not limited to, construction workers [1], underground coal miners [2], and tactical professionals, inclusive of military personnel [3], firefighters [4], and law enforcement [5]. Due to these potentially hazardous environments, persons working in these occupations wear specialised personal protective equipment (PPE). Law enforcement officers and military personnel wearing body armour [6] and firefighters wearing self-contained breathing apparatus (SCBA) and heat resistant clothing [7] serve as examples. One of the key and most common components of any occupational PPE is footwear.

Occupational footwear consists of a variety of boots (e.g., safety, combat, or work) [8,9] and shoes that are constructed from an assortment of materials [10] and worn in an occupational setting. Footwear construction can even vary within occupations depending on the task at hand. For example, military personnel have a standard issue boot constructed of leather but may also wear boots made of out of a cotton and nylon blend more suited to a humid environment [11]. Despite these variations, standards exist (e.g., Australian/New Zealand standard [12]]) to ensure proper protection of the worker. These standards necessitate certain design requirements (e.g., higher shaft height, toe caps, increased stiffness) to prevent slipping, contact with hazardous materials, or injuries due to external objects [13].

One potential downside to this focus on occupational safety is the detraction from functionality or comfort [13] which can lead to a safety risk. This consideration of workplace physical functional impact is crucial as occupational footwear provides an interface between a worker’s feet and the ground they work on [13]. The interaction between an individual’s feet and their working surface can influence essential physical movement requirements and when this interaction is negatively impacted (e.g., loss of ankle range of motion [ROM], peripheral neuropathy, etc.), a multitude of variables (like balance and gait [13,14,15]]) can be affected. For example, sole hardness in footwear, can impact a worker’s gait patterns [16,17]. Considering the potential impact of occupation footwear on physical function there is a need to ensure that the boots are designed to mitigate injuries, not to become a cause of them. For example, a leading cause of injury in military, firefighters, and law enforcement is through slips, trips, and falls [5,18,19]. Considering this, movement at the ankle (the ‘ankle strategy’) is an important part of balance recovery [20,21]. As such, actions which may impact on the ankle strategy, for example, boot upper stiffness [13], may inadvertently increase slip, trip, and fall injury risk through preventing the regaining of balance.

While research has shown that various types and constructs of footwear can impact aspects of physical function, the culmination of these effects of physical occupational task performance and musculoskeletal injury risk remains unknown. Further, the impact of specific differences in the construction of occupational footwear (e.g., materials used, shaft length, etc.) on these outcomes is yet to be realised. These are vital components of information that can affect the future production of occupational footwear, potentially resulting in safer and more functional footwear. As such, the aims of this scoping review were to gather, review, and synthesise the available literature on occupational footwear in relation to their impact on task performance and musculoskeletal injury risk with the intent to inform future tactical population footwear development. With this intent in mind, tactical occupations can work in various environments including offices, vehicles, and across various terrains [22,23,24]. As such, the scope of this review was deliberately kept broad to capture as many relevant research articles as possible.

## 2. Materials and Methods

A scoping review was conducted to identify and synthesise findings from published academic works that have investigated the impact of occupational footwear on physical task performance and musculoskeletal injury risk. The literature review protocol was guided by the Preferred Reporting Items for Systematic Reviews and Meta-Analyses extension for Scoping Reviews (PRISMA-ScR) [25].

### 2.1. Protocol and Registration

The project and the protocol for this scoping review were registered with the Open Science Framework registry on the 13 October 2021 (accessible at https://osf.io/3y2je) (Maupin et al. 2021).

### 2.2. Eligibility Criteria, Information Sources, and Search Terms

A systematic search of key databases was conducted in October 2021. These databases, PubMed, Scopus, ProQuest, Elton B Stevens Company (EBSCO) (inclusive of SPORTDiscus and Cumulative Index to Nursing and Health Care Literature [CINAHL]), and Web of Science were searched using search terms derived from two themes: ‘occupation’ and ‘footwear’. An example search string for the PubMed database can be found in Table 1 with the search strategies for the other databases detailed in Appendix A. Filters were utilised where available and included the following: published in English, published in the last 20 years, and human population. Identified studies were reviewed for duplicates and studies clearly not of relevance to this review (e.g., roller boots) were removed following which remaining studies were compared against specific inclusion and exclusion criteria (Table 2). At this point, excluded studies, and their reasons for exclusion were noted and are detailed in Appendix A.

### 2.3. Study Selection, Data Extraction, and Data Items

All articles identified in the initial search were imported into reference management software (Endnote X9, version X9.3.3, Clarivate Analytics, Philadelphia, United States), where duplicates were removed. Articles were then screened for relevance by title and abstract, with those that were clearly not relevant excluded at this stage. Remaining articles were obtained in full text for further assessment against the eligibility criteria to determine the final list of studies to inform the review. The screening process was conducted by one author (D.M.) and confirmed by a fellow author (R.P.) with consensus, where required, settled by a third author (R.O.). The results of the search, screening, and selection processes were documented in a PRISMA flow diagram [26]. Data pertaining to authors, occupational settings, footwear construction, impact on physical task performance or musculoskeletal injury risk, and key findings from each included study regarding the impact of the footwear, were extracted and tabulated. Data were extracted by one author (D.M.) and confirmed by fellow author (R.P.) with consensus, if required, settled by a third author (R.O.).

## 3. Results

The systematic search identified 19,614 records. After screening and selection procedures were undertaken, 50 studies were deemed eligible to inform this review. Figure 1 depicts the results of the systematic search and selection process, including a summary of reasons for exclusion of full text articles, which are further detailed in Appendix A. The 50 articles to inform the review included 25 quasi-experimental studies [2,7,8,9,10,27,28,29,30,31,32,33,34,35,36,37,38,39,40,41,42,43,44,45,46], eight cross-sectional [47,48,49,50,51,52,53,54], four randomised control trials [55,56,57,58], three cohort studies [59,60,61], three narrative reviews [62,63,64], three systematic reviews [13,65,66], three conference abstracts [67,68,69], and one case–control study [70].

The included articles represented work from 16 countries, with nine population types, and 12 footwear types reported on. The countries from which the articles were drawn included 21 from the United States of America [8,10,30,32,36,40,41,42,43,45,49,50,52,53,54,56,58,61,63,64,65]; seven from Australia [2,7,13,31,47,48,59]; three from Germany [9,33,44]; two from Brazil [38,39], England [29,62], India [37,70], Poland [34,60], and Saudi Arabia [27,28]; and one from Denmark [46], France [68], Japan [51], Netherlands [57], New Zealand [66], Portugal [67], South Korea [35], and Switzerland [55]. Regarding population types, 13 articles discussed military populations [8,13,37,38,39,40,41,44,50,63,64,65,66], nine discussed firefighters [7,10,13,30,32,35,42,43,60], six discussed trade/industrial workers (ship, construction, factory, and hairdressing) [8,33,45,67,68,70], six discussed mine workers [2,31,36,47,48,59], five discussed automotive workers [9,49,52,53,54], five discussed hospital/healthcare workers [29,51,54,57,58], two discussed healthy adults [34,46], two discussed university workers [27,28], two discussed school food service workers [56,61] and one review [62] discussed a broad range of populations. The types of footwear discussed included 19 articles on work safety boots/shoes (light, medium, heavy duty; industrial, comfort, unstable, gumboots, steel capped) [2,9,13,27,28,29,31,33,36,45,47,48,49,52,53,54,59,61,70], 13 on military boots (minimalist, standard) [8,13,37,38,39,40,41,44,50,63,64,65,66], 13 on sports shoes (trainers, running, tennis, basketball) [8,10,29,40,43,44,45,46,51,55,58,61,67], nine on firefighter boots (pull-up bunker, leather, rubber) [7,10,13,30,32,35,42,43,60], four on unstable shoes [49,58,67,68], three on sandals/sandal like conditions [29,34,35], two on clogs [29,57], two on dress shoes [29,44], two on hiking boots [13,34], and one on Wellington boots [29], and slip resistant shoes [56]. One review [62] discussed a broad range of shoe types. Three articles included a barefoot condition as a comparative condition [37,41,68].

Following the extraction and synthesis of the information from the collected articles, several emerging themes were identified in regard to occupation footwear and their impact on gait mechanics, joint ranges of motion, posture and balance, physiological impact (e.g., thermal, aerobic capacity, etc.), muscle activity, and occupational tasks. An overview of the findings in the included studies are detailed below. For full results, including statistical findings (e.g., measurements, *p*-values, confidence intervals, and test statistics) where applicable, please refer to Table 3.

### 3.1. Impact of Footwear on Occupational Performance

Occupational footwear has been found to impact on and during task performance. These impacts occur during gait [2,7,9,10,37,38,39,40,41,42,43,44,45,46] and on joint ranges of motion [9,34,36,42,43,44], posture and balance [32,45,70], physiological measures (like aerobic capacity, heart rates, thermal properties, etc.) [8,27,28,30,35,60], muscle activity [27,31,33,46], and occupational tasks (like running to an emergency department or lifting loads) [27,28,57]. The ranges of footwear types to be discussed below as having these occupational impacts include boots [2,7,10,32,34,36,37,38,39,40,41,42,43,44,45], safety shoes [9,27,28], athletic shoes [44,45], general work shoes [45,70], low cut work shoes [45], military dress shoes [44], rocker soled shoes [46], low cut enclosed sandal type footwear [34], and thin sandals [35].

### 3.2. Occupational Footwear and Their Impact on Gait Mechanics

Fourteen articles [2,7,9,10,37,38,39,40,41,42,43,44,45,46] investigated the impact of occupational footwear on gait mechanics; notably spatiotemporal parameters of gait (i.e., stride length, stride frequency/cadence, support phases, etc.), angular velocities, and biomechanical forces (e.g., ground reaction forces (GRF), plantar pressure, etc.). The range of footwear types included boots [2,7,10,37,38,39,40,41,42,43,44,45], athletic shoes [44,45], a low cut work shoe [45], a military dress shoe [44], safety shoes [9], and a rocker soled shoe [46].

### 3.3. Spatiotemporal and Angular Velocities of Gait

The majority of articles investigating the impacts of occupational footwear on spatiotemporal parameters and angular velocities of gait had participants wearing boots for military personnel [37,44], firefighters [10], underground coal miners [2], or construction workers [45]. Other footwear included athletic shoes [44,45], a low cut work shoe [45], and a rocker soled shoe [46].

In studies including military personnel, the wearing of occupational boots had various significant impacts on their spatiotemporal parameters of gait. Comparing barefoot versus standard issue military boots, Majumdar et al. [37] found that wearing military boots resulted in longer step and stride lengths (*p* < 0.001), a slower cadence (*p* < 0.001), less total and double support times (*p* < 0.05), a longer swing phase (*p* < 0.05), and increased single support (*p* < 0.01). All findings were presented bilaterally except for single support which was presented only on the right side. Additionally, it was found that military boots had no significant impact on step width or forward velocity [37]. Similar to these findings, Schulze et al. [44] found that military issued ‘combat’ boots (mass ± 1135 g) and military leather dress shoes (mass ± 530 g) resulted in significantly increased stride lengths when compared to a barefoot condition (*p* < 0.001 and *p* = 0.006, respectively). Furthermore, while stride lengths also increased in comparison to outdoor athletic shoes (mass ± 500–720 g) the increase was greater for the military boot (*p* = 0.005). During this study, the participating soldiers wore approximately 21 kg worth of load (including a 15 kg backpack). Of note, neither weight nor load distribution of the worn loads were found to influence stride length. As such, the increase in stride length could be attributed to the wearing of the boots.

Park et al. [10] compared rubber (±3.15 kg) and leather (±3.20 kg) firefighter boots to running shoes (±0.71 kg) to analyse the impact of occupational footwear on gait. Compared to running shoes, the leather and rubber boots resulted in greater anterior-posterior displacement and stride time, and decreased centre of pressure velocity [10]. The leather boots had greater time in stance and double support phases compared to running shoes and rubber boots [10].

Svenningsen et al. [46] compared a shoe with a rocker sole (MBT, model 1997) to a control shoe (REG, Nike Free, model 3.0) across participants from various occupations, including, but not limited to, car mechanics, casino employees, a teacher, a chef, and a nurse. Participants carried 1 L plastic bottles inside a backpack equating to 10% of their body weight. The results found no significant differences between the two shoe types in regard to stride duration or stride frequency.

Simeonov et al. [45] compared six footwear styles, three athletic (a running shoe (±312 g), tennis shoe (±416 g), and basketball shoe (±472 g)) and three occupational (a low-cut work shoe (±642 g), work boot (±690 g) and safety boot (±768 g)) styles and their impacts on gait across various simulated (via virtual reality) environments. The four environments consisted of walking on ground level, and three environments that simulated a rooftop (level 25 cm wide plank, 25 cm wide plank tilted to 14° and 15 cm wide plank) [45]. The study found that trunk angular velocity was similar at ground level, but high cut shoes resulted in significantly lower (*p* < 0.05) angular velocity while walking on planks, with safety boots [45]. This angular velocity was less affected by a smaller plank width when wearing the safety (*p* = 0.0516) or work boots (*p* = 0.0005) compared to the low cut work shoe [45]. On a 15 cm plank, the rearfoot angular velocity was significantly less in high-cut shoes than in low cut shoes (*p* < 0.005), but this was not seen when on a 25 cm plank [45].

### 3.4. Biomechanical Forces

The majority of articles investigating biomechanical force impacts of occupational footwear had participants wearing boots for military personnel [38,39,40,41], firefighters [7], and underground coal miners [2]. One study investigated the impacts of three different types of safety shoes on plantar pressure [9].

In a military population, GRF was measured between two military boot styles: one made with polyurethane (PU) weighing 380 g, and another made with styrene-butadiene rubber (SBR) [38] weighing 561 g. GRF was measured in both an unloaded and loaded (15 kg) condition. Muniz et al. [38] found that the PU had a significantly (*p* = 0.04) higher instantaneous loading rate when unloaded (26.1 ± 4.3% body weight per second [BW/s]) and when loaded (27.8 ± 4.1 %BW/s) compared to SBR boots (24.5 ± 6.6 %BW/s unloaded, 25.1 ± 5.2 %BW/s loaded) [38]. There was also a significantly higher (*p* < 0.01) median power frequency in the PU boots (24.6 ± 2.8 Hertz [Hz] unloaded, 24.9 ± 2.2 Hz loaded) compared to SBR boots (21.8 ± 3.3 Hz unloaded, 22.4 ± 4.7 Hz loaded) [38], suggesting higher forces throughout the duration of the measured stride. No significant differences were found in the first peak force [38]. Muniz and Bini [39] likewise compared GRF between three boots, two utilising SBR and varying rear foot thickness (Boot 1 30 mm weighing 632 g, Boot 2 20.6 mm weighing 530 g) and a PU boot weighing 423 g [39]. However, unlike the previous study, Muniz and Bini [39] did not find any significant differences in loading rate. A principal component analysis was conducted and found that Boot 1 had greater vertical scores (one component), and anteroposterior scores (one component) compared to Boots 2 and 3, but lower mediolateral scores (two components) compared to Boot 2 [39]. It was also found that Boot 3 had the highest comfort rating, but only significantly better than Boot 1 (*p* = 0.01) [39].

Neugebauer and Lafiandra [40] attempted to create a model to predict GRF in military soldiers by comparing military boots to athletic shoes across a variety of external loads (0 kg, 14 kg, 27 kg, and 46 kg) at various speeds (ranging from 1.5 to 3.6 mph at 0.3 mph increments with the exception of the 46 kg load maximised at 3.0 mph). The authors reported that the hardness of footwear was not a significant predictor of maximum GRFs (*p* < 0.70), but that the type of footwear was a significant factor in their model [40]. However, despite the type of footwear being a significant factor, it did not improve the predictive model enough to warrant inclusion in the predictive formula (*r*^2^ = 0.892 vs. *r*^2^ = 0.891) [40].

Oliver et al. [41] compared drop landings in military cadets wearing standard issue military boots, tennis shoes, and while barefoot. No significant differences in knee valgus were found between conditions, but it was found that tennis shoes had significantly (*p* < 0.05) higher ground reaction force as a percentage of body weight (1880 ± 379%) when compared to barefoot (1646 ± 359%) and boots (1833 ± 438%) [41]. These results differed in a population of firefighters. Vu et al. [7] compared landing results (without and with a load of 9.5 kg) in a cohort of firefighters wearing standard issue boots compared to personnel athletic footwear. Landing in firefighting boots resulted in significantly (*p* < 0.05) higher vertical reaction forces (2.40 times body weight) than athletic shoes (2.14 times body weight) [7].

Investigating plantar pressure and comfort in underground coal mining, Dobson et al. [2] compared four boot types consisting of combinations of stiff and flexible shafts with stiff and flexible soles (mass ± 0.94–0.98 kg). The results differed regarding contact area and peak pressure. A stiff boot shaft resulted in greater contact area and time under the medial (*p* < 0.001) and middle metatarsals (*p* = 0.016), while a flexible sole resulted in greater peak pressures under the medial metatarsals (*p* < 0.001) [2]. When asked about boot preference, coal miners were more likely to select one with a flexible shaft and stiff sole (*p* = 0.008) [2].

Oschman et al. [9] compared the effects of various occupational footwear on plantar pressure in automotive workers. Their study compared three types of safety shoes: Shoe 1 was a steel safety cap shoe weighing 530 g, Shoe 2 was an aluminium safety cap shoe weighing 630 g, and Shoe 3 was a steel safety cap shoe with a rocker bottom weighing 720 g. Shoe 1 had greater plantar pressure in the rearfoot and one forefoot zone compared to Shoe 2 and Shoe 3, but less plantar pressure than the other two shoes throughout the midfoot [9]. Shoe 3 tended to have more plantar pressure in the rear and midfoot compared to Shoe 2 and less pressure in the forefoot [9]. Shoe 3, with the a rocker bottom, had a smaller centre of pressure (COP) range in the posterior-anterior direction compared to Shoes 1 and 2 [9]. Likewise, Shoe 3 had a similar 50th percentile COP in the medial lateral direction as Shoe 1, both of which were significantly less than Shoe 2 (*p* = 0.002, *p* < 0.001 respectively) [9]. Shoe 3 also had significantly less COP range in the medial-lateral direction as Shoe 1 (*p* = 0.022) and Shoe 2 (*p* = 0.001) [9].

### 3.5. Impact of Occupational Footwear on Joint ROM

Six articles [9,34,36,42,43,44] investigated the impacts of occupational footwear on the ROM across various joints (e.g., hip, knee, and ankle). The range of footwear types including boots [34,36,42,43,44], athletic shoes [44], low cut enclosed sandal type footwear [34], and safety shoes (steel safety cap, aluminium safety cap, and steel safety cap with rocker-bottom) [9]. The participants wore boots for the military personnel [44], firefighters [42,43], automotive workers [9], and general occupational population [34,36].

Schulze et al. [44] examined the impacts of military issued ‘combat’ boots (mass ± 1135 g), military leather dress shoes (mass ± 530 g), two types of outdoor athletic shoes (mass ± 500 and 720 g respectively) and indoor athletic shoes (mass ± 600 g) versus a barefoot control condition on joint ROM. At the knee, the different footwear was purported not to lead to a change in knee ROM in flexion-extension (i.e., total range across the knee), but resulted in a shift to greater ranges of knee flexion with fewer ranges of knee extension. This was also found when comparing the unloaded to the loaded (21 kg) condition (*p* < 0.043). At the ankle, a significant reduction in plantar flexion was found when boots were compared to barefoot (*p* < 0.001) and other footwear (*p* < 0.05) conditions. Still at the ankle, the equipment load was not found to influence ROM. No significant differences in hip ROM were found [44] until comparing loads which resulted in a reduction of extension and a significant increase in flexion (*p* = 0.005).

Park and associates analysed the impact of various firefighter boots on ROM in the lower limb [42,43]. In one study, comparisons were made between running shoes (mass ± 0.71 kg) and rubber boots (mass 3.15 kg) [43]. It was found that the rubber boots had significant impacts on the ROM throughout various lower limb joints during the gait cycle. For example, in the sagittal plane wearing rubber boots resulted in significantly more hip flexion (*p* < 0.001), knee flexion (*p* < 0.001), and less dorsiflexion (*p* < 0.001) and great toe flexion (*p* < 0.001) [43]. Wearing the rubber boots also resulted in greater inversion and rotation at the ankle (*p* < 0.026, and < 0.001 respectively) as well as adduction and rotation at the ball of the foot (*p* < 0.001 for both) [43]. Park et al. [42] also compared the impacts of three various boot (leather) heights (25.4 cm, 30.48 cm, and 35.56 cm) on ROM during walking and occupational tasks (duckwalking and ladder ascent/descent) [42]. Overall, this study found significantly less ROM in hip, knee, and ankle during gait with higher boots [42]. These ROM decrements were also found while completing various tasks [42]. For example, while duckwalking firefighters in higher boots had significantly lower knee (*p* = 0.002) and hip ROM (*p* = 0.006) when accounting for knee height. When accounting for leg length, firefighters with taller knee height had significantly smaller ankle ROM in high boots than low boots during ladder ascension (*p* = 0.025).

Irmańska and Tokarski [34] compared two types of boots (protective ankle boots and low-cut sandal-like protective footwear) in a cohort of young and old which included firefighters, drivers, and farmers (younger group) and farmers and security personnel (older group). Young workers were more likely to have greater talocrural ROM compared to their older counterparts [34]. The protective ankle boots only differed from the sandal-like footwear in left talocrural ROM during stair climbing for the young group [34]. As such, the ROM available at the ankle may meet a diminishing impact based on the age of the wearer (i.e., the older wearer with less ROM at the ankle is less impacted).

Kocher et al. [36] recruited workers from the National Institute of Occupational Safety and Health (specific industries were not provided). Authors of this study compared four boot styles: (1) hiking style boot with steel toe, (2) hiking style boot with steel toe and metatarsal guard, (3) wader style boot with steel toe, and (4) wader style boot with steel toe and metatarsal guard [36] while participants walked up and down ramps at up to 20° inclination/declination. The authors found that boot style had significant interactions with hip and knee ROM while descending (hip L, *p* = 0.005, hip R, *p* = 0.01, knee L, *p* = 0.004, knee R, *p* = 0.005), and ankle ROM while ascending (L, *p* < 0.001; R, *p* < 0.001) and descending (L, *p* < 0.001; R, *p* < 0.001) [36]. Regarding ROM, wader style boots consistently had less ROM than hiker style boots, except ascending hip ROM. Specifically, the wader style boots with the metatarsal guard were the most restrictive [36]. These restrictions were especially prominent at the ankle (ascending and descending) and at the knee when descending [36]. The wader style boots only had one significant difference between the two which was descending left hip ROM with the steel toe cap only configuration demonstrating greater ROM [36].

Oschman et al. [9] also compared the effects of various occupational footwear on ROM. Their study compared three types of safety shoes: Shoe 1 was a steel safety cap shoe, Shoe 2 was an aluminium safety cap shoe, and Shoe 3 was a steel safety cap shoe with rocker-bottom [9]. The results of their study found that Shoe 1 had significantly greater trunk inclination (measured by 50th percentile) during gait compared to Shoe 2 (*p* = 0.005) and Shoe 3 (*p* < 0.001). Shoe 3 had significantly less hip flexion (measured by 50th percentile) than both Shoe 1 (*p* = 0.001) and Shoe 2 (*p* = 0.046). Lastly, when participants were wearing Shoe 2, they moved through a greater knee ROM (measured as a difference between 95th and 5th percentiles) when compared to Shoe 1 (*p* = 0.008) and Shoe 3 (*p* < 0.001) [9].

### 3.6. Impact of Occupational Footwear on Posture and Balance

Three [32,45,70] articles reported on the impacts of occupational footwear on the wearer’s posture and balance. The range of footwear types included boots [32,45], athletic shoes [45], and general work shoes [45,70]. The participants included firefighters [32], hairdressers [70], and construction workers [45]. All three papers did, however, have a different focus. Only one study investigated the nature of footwear on posture and balance directly [32]. The remaining two studies investigated either the impacts of wearing a specific type of footwear over an 8-week period on posture and balance [70], or reported subjective responses from wearers in relation to their perceptions of stability while wearing one of six different types of footwear [45].

In a population of male firefighters, Garner et al. [32] compared the difference between rubber (~2.93 kg) and leather (~2.44 kg) firefighter boots during a Simulated Fire Stair Climb (SFSC) task conducted on a Stairmaster stepmill where firefighters stepped at a rate of 60 steps per minute wearing 34 kg of PPE. The results from their study suggested that rubber boots resulted in greater sway (anteroposterior and medial-lateral) parameters. The authors suggest that, based on the significant main effect and interactions, the leather boots maintained postural stability better than the rubber boots and that rubber boots may pose a postural risk for firefighters [32].

Sousa et al. [70] further compared the effects of unstable shoes (worn for 8 weeks) on postural control in a population of hairdressers. A control group wore their regular shoes while the intervention group wore the unstable shoes for the duration. The authors analysed various measures of balance and postural control including muscle activation, centre of pressure, and body oscillation, finding the unstable shoes resulted in a greater demand of postural control compared to a barefoot condition—even after the 8-week training period [70]. However, it was also noted that following the 8-week intervention period, those who were wearing the unstable shoes had more efficient and effective postural control when compared to the control group who wore the unstable shoes during the assessments but without the prolonged exposure [70].

Simeonov et al. [45] compared six footwear styles, three athletic (a running shoe (±312 g), tennis shoe (±416 g), and basketball shoe (±472 g)) and three occupational (a low-cut work shoe (±642 g), work boot (±690 g) and safety boot (±768 g)) styles. Of relevance to balance, the authors reported that the perceptions of instability among workers were similar regardless of the shoe when on ground level, but when simulating walking at roof height workers felt significantly (*p* < 0.05) more stable in high cut work shoes.

### 3.7. Impact of Occupational Footwear on Physiological Outcomes

Five [8,27,28,30,35] articles reported on the impacts of occupational footwear on various physiological measures (e.g., thermal, aerobic capacity/VO_2_ (energy cost), etc.). The range of footwear types included military footwear [8], firefighter boots [30,35], thin sandals [35], and leather safety shoes [27,28], across military [8], firefighter [30,35] and university worker [27,28] populations.

Pace et al. [8] compared the use of a minimalist style boot (Tactical Research MiniMil Ultra-Light Minimalist Tactical Military Boot TR101 weighing 500 g per boot) to standard issue military boots (Belleville 310 T Hot Weather Standard Tactical Boot weighing 801 g per boot). Both boots had shaft heights of 20.3 cm and 3–4 mm sole tread depth. Participants (including military and law enforcement personnel) completed four, 5 min treadmill exercise bouts with a 16 kg load at 5.75 km/h followed by two more 5 min bouts at a running intensity of 7.40 km/h. The authors reported significantly better respiratory exchange ratios (*p* < 0.01, Cohen’s *d* = 0.90), and decreased energy cost requirements (*p* < 0.05, Cohen’s *d* = 0.31) when participants ran in the minimalist boots. These improvements did not exist during the walking trials. This better performance in minimalist boots was also associated with decreased ratings of perceived breathing exertion when walking and running (*p* < 0.01, Cohen’s *d* = 0.43 and *p* < 0.05, Cohen’s *d* = 0.33, respectively), as well as decreased ratings of perceived lower limb exertion while running (*p* < 0.05, Cohen’s *d* = 0.69) [8].

Chiou [30] studied the physiological impact of firefighter boots on the wearer, comparing four boots of various mass and materials. All four boots met the National Fire Protection Association standards for structural firefighting and were pull-up bunker boots that were commercially available. The boots were: a hybrid upper with a combination of leather and fabric upper and less flexible soles weighing 2.05–2.48 kg, a leather upper with less flexible soles weighing 2.46–2.93 kg, a leather upper with more flexible soles weighing 2.56–3.10 kg, and a rubber upper with more flexible soles weighing 3.36–3.82 kg. It was found that, in general, heavier boot mass resulted in higher oxygen consumption (VO_2_), relative VO_2_, and VCO_2_ in wearers, suggesting a greater energy cost imparted by heavier boots. Conversely, there were significant decreases in VO_2_ for more flexible soles. The significant effects of boot mass and sole flexibility (*p* < 0.05) were consistent for both the male (n = 14) and female (n = 13) firefighter participants.

Furthermore, in a firefighter population, Lee et al. [35] assessed the impact of various parts of firefighter equipment, including boot wear versus no boot wear (wearing thin sandals), on heat production. Following a protocol whereby each trial began with a 10-min seated rest followed by 30 min on a treadmill (5.5 km/h, 1% slope) and a 20-min seated recovery. When firefighters were not wearing their boots (but wearing other PPE such as helmet, gloves, or a breathing apparatus), there were similar results in sweat rate, rectal temperature, skin temperature, and heart rate, when compared to the no thermal clothing and no equipment conditions [35]. Likewise, at the end of the treadmill exercise, VO_2_ impacts were smaller in no-boots trial than in any of the other boot conditions. As a result, the no-boots, no-equipment and no-thermal-clothing ranked as least thermal strenuous conditions. Thus, the authors concluded that wearing boots led to greater heat production as the distance of the foot from the body’s centre of mass makes heat distribution mechanically inefficient.

Al-ashaik et al. [27] compared light regular (mass = 0.9 kg, low cut, rubber sole), medium (mass = 1.05 kg, low cut, PU moulded sole) and heavy-duty (mass = 1.45 kg, high cut, PU sole) leather shoes across a group of university workers performing lifting tasks. Both working heart rates and incremental aural-canal temperature for participants wearing the heavy-duty safety shoes were higher than their working heart rates and temperature while wearing either the medium-duty or light safety shoes (*p* < 0.001) [27]. Alferdaws et al. [28] compared the same shoes in a similar population but found no significant differences in respiration rate, minute ventilation, or relative VO_2_ [28]. Light-duty shoes did result in significantly less discomfort than heavy-duty (*p* < 0.001) and medium-duty (*p* < 0.001) shoes [28].

### 3.8. Impacts of Occupational Footwear on Muscle Activity

Four studies [27,31,33,46] investigated the impacts of occupational footwear on muscle activity. The studies compared muscle activity across boot [31] and shoe types (based on shoe mass [27] or stability [46]) and level of cushioning [33].

Huebner et al. [33] compared various levels of heel cushioning in safety shoes previously worn by subjects and a test shoe with an interchangeable cushioning element in canteen workers. The impact of the cushioning in the test shoe was investigated through three conditions, being no cushioning (dummy insert), optimal cushioning (recommended), and too soft cushioning. Optimal cushioning was associated with one of four recommended body weight categories (provided by the manufacturer) being <57 kg, 58–79 kg, 80–91 kg, and >91 kg. The authors found that optimal cushioning was associated with decreased muscle activity per distance travelled (an indirect measure of energy expenditure), though this only occurred in the back musculature. Optimal cushioning also resulted in greater scores for the normalised mean range for leg muscles at preferred and fast walking speeds [33].

In the study by Al-Ashaik et al. [27] comparing light regular (mass = 0.9 kg), medium (mass = 1.05 kg,) and heavy-duty (mass = 1.45 kg,) leather shoes across a group of university workers performing lifting tasks, muscle activation for the biceps brachii, trapezius, anterior deltoid, and erector spinae were captured. For the biceps brachii, the percentage of a maximal voluntary contraction (%MVC) was significantly lower with the light regular safety shoes as compared to the heavy-duty safety shoes (*p* < 0.04) when performing 1 lift/minute in a temperature of 20 °C (Wet Bulb Globe Temperature (WBGT)). In addition, %MVC was again significantly lower wearing the medium safety shoes than when wearing the heavy safety shoes (*p* < 0.01) when performing 5 lifts/minute. However, in an environmental temperature of 30 °C (WBGT), the %MVC of the biceps brachii muscle was significantly lower while performing 1 lift/minute wearing the medium safety shoes than when wearing either the light (*p* < 0.002) or heavy-duty safety shoes (*p* < 0.022). The only other muscle group to interact with shoe type was the trapezius muscle, presenting with significantly lower %MVC when lifting while wearing light versus heavy safety shoes (*p* < 0.04). No interactions were found in %MVC for the anterior deltoid or erector spinae muscles.

Svenningsen et al. [46] compared an unstable shoe (shoe with a rocker sole) to a control shoe across participants from various occupations wearing backpacks. Participants walked on a treadmill at their preferred speed for two minutes in either shoe condition and in an unloaded or loaded (10% of their body weight) condition. The authors found that wearing unstable shoes resulted in significantly higher muscle activity in the longissimus thoracis (the largest erector spinae muscle) and iliocostalis muscles (immediately lateral to the longissimus) of the back when the electromyography (EMG) was expressed as the root mean squared (RMS). This result occurred both without and with load (*p* < 0.05). However, only the longissimus thoracis was significantly higher in the unstable shoe type when expressed as peak EMG.

Dobson et al. [31] compared four boot types (flexible shaft with stiff or flexible sole, and stiff shaft with stiff or flexible sole, mass ± 0.94–0.98 kg) and their impact on muscle activity (vastus lateralis, semitendinosus, tibialis anterior, peroneus longus, and gastrocnemius medialis) in participants who worked as underground coal miners or trade workers. The type of sole and shaft utilised did have significant impacts on various muscles and their peak activity and onsets [31]. This further varied depending on whether the participant was walking on gravel or soft surfaces [31]. It was noted that boots that were either all flexible or all stiff were likely to have higher muscle activity and earlier onsets, potentially leading to higher rates of fatigue, therefore suggesting a mixed construction may be a better option [31].

### 3.9. Impact of Occupational Footwear on Occupational Tasks

Only three studies were found to examine the direct impacts of occupational footwear on an occupational task [27,28,57]. In a running task, Elbers et al. [57] compared responses to an emergency call in a population of intensive care medical professionals wearing either small or large clogs. It was found that smaller size clogs resulted in a quicker response time when running 125 m from a coffee break room to the emergency department elevator. These results remained extant even when accounting for gender, age, height, own shoe size, or fitness [57]. Further, there no significant differences in adverse events (e.g., lost pocket items, pain when running, etc.) or comforts between the two sizes [57].

In a lifting task, the aforementioned study by Al-Ashaik et al. [27] compared light, medium, and heavy-duty leather safety shoes across a group of university workers. The authors found that the lighter leather shoes resulted in a higher mean weight lifted, significantly higher maximum acceptable weight of lift (MAWL) (*p* = 0.013), and significantly lower rating of perceived exertion (*p* < 0.01). These results were supported by Alferdaws et al. [28], comparing the same shoes in a similar population, who likewise found that MAWL was higher while wearing light duty shoes (*p* < 0.041).

### 3.10. Miscellaneous

Irmańska [60] compared three boots (novel liner with ventilation, novel liner with no ventilation, wool liner) in a cohort of firefighters. The average mass of a pair of boots with liners included was 3.5 kg. The firefighter participants were required to walk at a speed of 4–5 km/h for 5 min, ascend and descend stairs (17 ± 3 steps) for a maximum of 1 min, before kneeling/crouching for 1 min. Thermal (*p* < 0.05) and moisture (*p* < 0.01) sensations were more strongly reported by the participants while utilising the wool liner, but there were no significant differences between groups with walking or kneeling/crouching movements [60].

### 3.11. Impact of Occupational Footwear on Musculoskeletal Injury Risk

A total of 16 studies reported on the occupational footwear worn at the time of injury [29,47,48,49,50,51,52,53,54,59] or potential injuries/injury risks (e.g., tripping) [30,55,56,58,60,61] associated with footwear types. Populations included Army Reserve Officers [50], automotive workers [49,52,53,54], hospital workers [29,51,55,58], coal miners [47,48,59], firefighters [30,60], school district workers [56], and ship workers [61].

#### 3.11.1. Occupational Footwear Worn at the Time of Injury

The occupational footwear worn at the time of injury was reported in ten studies [29,47,48,49,50,51,52,53,54,59]. Occupational populations included, Army Reserve Officers [50], automotive workers [49,52,53,54], nurses [29,51], and coal miners [47,48,59].

Scott and colleagues [50] surveyed injured recruits and the type of footwear worn at the time of injury. Of the 41 injuries recorded, seven injuries occurred when the recruits were wearing government issued boots, 17 while wearing conventional running shoes, and 17 while wearing ‘other’ types of footwear [50]. The results of a chi squared test showed that boot type (i.e., government issued boot and government-approved boot) was not significantly associated with injury (χ^2^ = 0.19, *p* = 0.91) [50].

Werner and colleagues [52,53,54] conducted three studies investigating various lower limb injuries (plantar fasciitis, hip pain, and foot and ankle disorders) in automotive workers. No significant associations between firmness of heel, insole, or shoe rotation and incidence of hip disorders were found [53]. Similarly, outer sole stiffness was not significantly associated with foot and ankle disorders but was associated with new onset foot and ankle disorders (middle tertile stiffness OR 8.2, 95% CI 1.01–65.6; upper tertile stiffness OR 18.9, 95% CI 2.2–165.8) [54]. Outer sole stiffness was likewise not associated with plantar fasciitis in this population, but workers who rotated shoes during the work week presented with reduced risk of plantar fasciitis (OR 0.30, 95% CI 0.1–0.7) [52]. Gell et al. [49] also compared the differences in footwear in automotive workers finding that there were significant differences in the footwear of workers who complained of lower limb fatigue at the end of the day. While limited survey data were available, the results suggest that workers who wore shoes with inserts or harder outsoles were more likely to report lower limb fatigue [49].

Tojo et al. [51] conducted a cross-sectional study of 636 nurses and reported that low shoe comfort score was associated with reported presence of foot and ankle pain (OR 2.12, 95% CI 1.13–3.50, *p* = 0.002), and presence of foot pain (OR 1.78, 95% CI 1.12–2.69, *p* = 0.006) and disabling foot pain (OR 1.76, 95% CI 1.01–3.11, *p* = 0.04) as measured by the Manchester Foot Pain and Disability Index. A medium comfort score was significantly associated with the presence of any foot or ankle pain [51]. Likewise a study by Anderson et al. [29] compared shoe comfort and pain in a population of nurses finding that greater footwear comfort lead to a decreased risk of hip/thigh pain (OR = 0.9, 95% CI 0.7–1.0), knee pain (OR = 0.9, 95% CI 0.7–0.9), and foot pain (OR = 0.8, 95% CI 0.7–0.9).

Dobson and colleagues conducted a series of studies analysing underground coal miners and their footwear [47,48,59]. In one study, the authors found that lower back, hip, ankle, and foot pain were significantly related to certain design characteristics (such as heel breadth, ball of foot girth, instep height, and toe angle, respectively) [59]. A second study by this research group analysed the impact of gumboot and leather lace-up boots finding no significant difference between the two in terms of presence of lower back, hip, knee, ankle, or foot pain [47]. Boot type was related to the type of foot condition workers were likely to report with gumboot wearers more likely to report ball of foot pain (*p* = 0.002), lateral malleolus pain (*p* = 0.040), and arch pain (*p* = 0.035) while leather boot wearers were more likely to report having corns (*p* = 0.034), navicular pain (*p* = 0.029), bunions (*p* = 0.035), sole pain (*p* = 0.006), heel pain (*p* = 0.028), and cuboid pain (*p* = 0.001) [47]. Additionally, it was found that participants with hip pain were more likely to rate their work boot fit as very poor, poor, or reasonable (*p* < 0.05), while those with foot pain were more likely to rate their boots as uncomfortable to indifferent (*p* < 0.001) [48].

#### 3.11.2. Occupational Footwear and Injury Risk

The potential for occupational footwear to directly cause injuries [61], chaffing and discomfort [60], pain and disability [55,58], as well as mechanisms associated with causing injuries (like tripping [30] and slipping [56]) were investigated in firefighters [30,60], school district workers [56], hospital workers [55,58], and ship workers [61].

Talley et al. [61] aimed to calculate the probability of ship workers suffering an injury while at sea with the wearing of safety boots serving as a predictive factor. Of injured individuals, 42.1% suffered an injury while wearing a safety boot, but the use of safety boots was found to decrease the probability of injury on only one type of container ship (noted for having different union officers) [61]. The type of injuries were not specified so the potential role of safety boots in reducing either musculoskeletal injuries or workplace accident injuries is unclear [61]. Irmańska [60] compared three different types of liners (with ventilation, no ventilation, and standard wool) placed inside firefighters boots. The study found that using boots with wool liners were significantly more likely to result in complaints of chaffing (*p* < 0.05) and higher discomfort (*p* < 0.05).

Viera et al. [58], conducted a randomised control trial on the effects of unstable (rocker bottom) shoes versus regular occupation footwear in nurses. The use of unstable shoes resulted in significantly lower levels of pain at Week 4 (*p* = 0.016) and 6 (*p* < 0.001), as well as significantly reduced reports of disability at Week 6 (*p* = 0.020) [58]. Similarly, Arman et al. [55], examining the impact of unstable shoes via a randomised control trial of hospital workers, found a significant decreases in pain (*p* = 0.001 to 0.037) and a decrease, though not significant, in disability after a period of 6-weeks.

The injury mechanism of trips and slips were investigated in two studies [30,56]. Chiou et al. [30] studied how various firefighter boots may lead to tripping when navigating obstacles. The authors reported that heavier boots resulted in decreased trailing leg clearance (2.9–4.4 cm less clearance for every 1 kg increase for low and high obstacles (*p* < 0.003, *p*< 0.02 respectively), decreased distance away from obstacles before climbing over (*p* < 0.05), and increased lead heel contact velocity (*p* < 0.02) [30]. Bell et al. [56] examined the impact of providing, rather than recommending, slip resistant footwear for school district workers. The use of slip resistance footwear significantly decreased the probability of suffering a slipping injury (OR_adj_ = 0.33, 95% CI 0.17–0.63) [56].

### 3.12. Systematic and Narrative Reviews

Included in this scoping review, three narrative reviews [62,63,64], and three systematic reviews [13,65,66] were identified and key findings were noted.

Dobson et al. [13] completed a systematic review assessing the impact of occupational footwear on gait, noting that 18 studies met their specific criteria. A cross-over between included studies of the review by Dobson et al. [13] and this current review is present, so specific results are only briefly mentioned. Dobson et al. [13] reported three main boot design features that impact gait, being shaft height, shaft stiffness, and boot mass, with boot mass being the most variable. Shaft height and stiffness were found to reduce ROM, particularly at the ankle and foot, potentially forcing workers to rely more heavily on other proximal joints, such as the hip [13]. However, a higher shaft height was reported to improve ankle stability and reduce the number of ankle injuries experienced by a population [13]. Boot mass likewise influenced certain parameters of gait with heavier boots increasing energy expenditure, heel contact velocities, and reducing trailing limb toe clearances.

Chander et al. [63] performed a narrative review discussing the potential benefits of a minimalist style boot in military personnel. The authors noted that construction elements of a boot (such as a lower heel height, thin and firm midsole, and lower mass) could improve postural stability, proprioception, and energy expenditure [63]. This review also self-cites conducted research by the authors noting that a minimalist boot performed better in reducing slip-induced falls, improved static and dynamic balance, and decreased muscular exertion—though these studies were conducted in the general population [63]. In their narrative review, Anderson et al. [62] noted that there are clear limitations in this area of research, including a lack of methodological standardisation due in part to a lack of detail in methodological reporting and a range of techniques used to measure the same variables. Their review also noted that a thin sole in nursing shoes was likely to increase the number of discomfort complaints for the back and lower limb, and that harder footwear increased the risk of lower extremity self-reported fatigue (RR = 2.6, 95% CI 1.3–5.3).

Yueng et al. [66] conducted a systematic review, reviewing 25 trials across a range of injury prevention topics. Two studies in their review compared standard issue leather military boots to “tropical” combat boots made with a cotton and nylon bend. No significant difference was found between the two boot types and soft-tissue injuries in the lower limbs [66]. Knapik et al. [65] conducted a systematic review of 16 studies comparing injuries before and after 1982 in the military, with 1982 signifying a change from boots to running shoes for physical training. The results of this study were then summarised and placed into historical context in a narrative review by the same lead author [64]. Overall the results of a χ^2^ analysis and meta-analysis showed no significant differences in injury incidence between the two time periods [65]. This result holds true even when comparing within sexes, and comparing overall injury incidence and lower extremity injury incidence [65].

### 3.13. Conference Abstracts

Three [67,68,69] conference abstracts were identified and included in this scoping review. Chorsiya et al. [67] published a conference abstract noting that shoe characteristics (such as toe cap, sole of shoe, mass of shoe) had significant interactions on the centre of pressure displacement, though further information and statistical results could not be found. Choukou and colleagues [68,69] likewise published two conference abstracts regarding unstable shoes versus barefoot, standard safety shoes, more comfortable safety shoes, and a safety shoe with a convex sole (unstable) meant to improve ergonomics. It was noted that the unstable shoes resulted in significantly higher pressures (anteroposterior magnitude, total area, length of centre of pressure, and velocity of centre of pressure (*p* < 0.05)) as compared to the other footwear [69]. The authors also found that the unstable shoes, resulted in greater toe off peak force (*p* < 0.05), and heel strike peak force (*p* < 0.05) with no difference in midstance peak force [68]. 

## 4. Discussion

The aims of this scoping review were to gather, review, and synthesise the available literature regarding the impact of occupational footwear on task performance and musculoskeletal injury risk to inform footwear design for tactical occupations. From an initial 19,614 identified articles, 50 articles met the criteria for review representing a wide range of occupational footwear types including boots, safety shoes, athletic shoes, general and low-cut work shoes, military dress shoes, rocker soled shoes, low cut enclosed sandal type footwear, and thin sandals. Generally, occupational footwear was found to impact gait and angular velocities, joint ranges of motion, posture and balance, physiological measures, muscle activity, and occupational tasks. Occupational footwear associated with injuries included boots, conventional running shoes, shoes with inserts, harder/stiffer outsoles or thin soles, and shoes with low comfort scores; although the findings were mixed. Occupational footwear was also linked to potentially causing injuries directly as well as leading to mechanisms associated with causing injuries.

### 4.1. Task Performance

The volume of evidence from this review suggests that occupational footwear can have significant impacts on physical task performance. The occupational footwear informing this research was typically found to impact gait by increasing step and stride length [37,44] slowing cadence [37], and decreasing support time [37]. More specifically, boots were found to increase stride lengths to a greater extent than athletic shoes [44], while leather boots led to greater time spent in the stance and double support phases of gait when compared to rubber boots and running shoes [10]. Factors that can increase stride length warrant consideration. For example, Pope et al. [71] found that forced increases in stride lengths were associated with pelvic stress fractures in female soldiers when undertaking formation load carriage sessions in boots.

GRFs were found to differ between two boot types in both an unloaded and a loaded (15 kg) condition with the PU boots recording higher instantaneous loading rates when compared to heavier (+181 g or +67% heavier) SBR boots potentially due to better cushioning in the SBR boots [38]. As such, even though PU boots weighed less, and presented with increased energy absorption and lower hardness, the offsets were insufficient to reduce impact. However, these results may not always be consistent as additional research by the same lead author [39] of the above study found no difference in GRF between the two boot types. A potential reason could be the findings of Neugebauer and Lafiandra [40] who noted that the hardness of footwear was not a significant predictor of maximum GRFs even though the type of footwear worn (military boots versus athletic shoes) was a significant factor in their model.

Overall, it appears that the use of occupational footwear results in higher GRF and greater plantar pressure. Factors contributing to GRF are of note given that GRF is associated with running injuries [72] and that running is a leading cause of injuries in military personnel [73,74]. Furthermore, increases in GRF also extend to the lumbopelvic region, with research by Vu et al. [7] finding various forces throughout the lumbopelvic region being significantly higher in firefighters wearing these boots. As such, increased GRFs impart increased load to the human body just as they increase energy expenditure [75,76]. As such increased GRFs may impact injury risk [77] and increase the workload requirements of everyday activities [75].

The evidence also noted the potential for occupational footwear, boots typically worn by tactical personnel, to negatively impact joint ROM, specifically at the ankle joint [42,43,44]. Greater boot shaft height and stiffness were found to reduce ROM, particularly at the ankle and foot, potentially forcing workers to rely more heavily on more proximal joints, such as the hip [13]. In a population of firefighters, rubber boots were found to significantly impact joint ROM throughout various lower limb joints during the gait cycle when compared to running shoes with a significant reduction in ankle dorsiflexion. Likewise, participants wearing wader style boots were consistently found to have less joint ROM than hiker style boots [36]. In an occupational task context, boot styles have been found to reduce ankle ROM during tasks like ascending ladders [42] and ascending and descending walkways [36]. Conversely, a higher shaft height has been reported to improve ankle stability and reduce the number of ankle injuries experienced by a population of Royal Marine recruits [13]. Whether or not the incidence of injuries higher up the kinetic chain (e.g., the knee) changed (e.g., increased) is not known as the original research by Riddell [78] could not be sourced. Of note, the work of Irzmańska and Tokarski [34], on the impacts of occupational boots on ankle ROM, suggests a diminishing impact of ankle ROM loss based on the age of the wearer whereby ROM loss may be greater in younger as opposed to older wearers (due to age related loss of ROM at the ankle).

Considering the aforementioned findings, loss of ROM at the ankle is a notable concern as the ankle (and its ROM) forms an integral part of the human body’s balance strategy. When reacting to an unexpected external force, the initial balance response is an ankle strategy, followed by a hip strategy, and finally a stepping strategy [20,21]. If ankle weight bearing dorsiflexion ROM is limited, the ability to maintain balance is impacted. This in turn could lead to an increased risk of a slip, trip and fall [42]. Noting that slips, trips, and falls are leading workplace injuries, notable so in military, firefighter, and law enforcement populations [5,18,19], occupational footwear that further increases this risk is of concern. Furthermore, limitations in ankle joint ROM may influence ROM requirements at more proximal joints [13,42,43]. This supposition is supported by the findings of Park et al. [43] who found increases in the knee and hip ROM with concomitant decreases in ankle ROM when firefighters wore rubber boots. Thus, more proximal joints and muscles supporting those joints could be exposed to greater workloads. Noting that muscle stressing is likewise a leading source of injury in tactical populations [18] the second order effects of reduced ankle ROM need to be considered (e.g., reduced ankle range, increasing the knee ROM requirement, increasing thigh muscle workload, leading to thigh muscle stressing).

There are, however, conflicting results regarding the impacts of occupational footwear on joint ROM at the knee and hip joints. Two studies found that occupational footwear resulted in greater flexion at the knee and hip, possibly to overcome the lack of ankle ROM [42,43] while a study by Kocher et al. [36] found decreases in knee ROM. These differences are potentially due to the boots worn in the study by Kocher et al. [36] which were slightly rolled down to allow for placement of gait analysing markers. When compared to being barefoot, Irmańska and Tokarski [34] found that footwear impacted ROM with low cut boots resulting in less knee and ankle joint ROM [34]. Apart from the type of footwear used in the studies, another potential reason for these differences lies in how ROM was measured. For example, in their study, Schulze et al. [44] found that different footwear did not to lead to a change in knee ROM in flexion-extension (i.e., total range across the knee), but resulted in a shift to greater ranges of knee flexion with fewer ranges of knee extension. Thus, knee ROM can be said to increase (if only looking at flexion) or remain the same (if considering the range between flexion and extension).

Little research was done on posture and balance. Firefighters performing an SFSC task were found to have an increased postural sway when wearing rubber (as opposed to leather) boots to the extent that the authors proposed that rubber boots may pose a postural risk for firefighters [32]. The protocol employed in this study by Garner et al. [32] assessed balance over 60 sec but without any load. The addition of load may exacerbate these impacts. A study comparing different types of body armour in police officers (2.1 to 6.4 kg) found that, when wearing body armour (regardless of type), postural sway increased significantly over 30 sec [6]. Given that police officers (and military personnel) may be required to stand in place for a period of time (e.g., vital asset protection) while wearing occupational loads (law enforcement 10 [79]–22kg [80]; military personnel 45+ kg) [80]), boots that increase postural sway may potentially exacerbate sway and as such balance (and energy) requirements associated with occupational load carriage.

Occupational footwear may, however, increase postural control. In a study comparing rocker style (unstable) shoes versus convention shoes, participants wearing the unstable shoes were found to demonstrate more efficient and effective postural control when compared to the control group following an 8-week intervention period [70]. In addition, in a study by Simeonov et al. [45] comparing athletic, work shoe, work boot, and safety boots the perceptions of instability among workers was similar regardless of shoe type when on ground level, but workers felt significantly more stable in high cut work shoes when simulating walking at roof height. These results suggest a potential learning and adaptability effect which should be noted in research undertaken with different occupational footwear and that the context in which research takes place could influence subjective feedback from participants.

Occupational footwear mass was found to have a notable impact on various physiological measures. In their study, Lee et al. [35] found that firefighter boots had the biggest impact on energy expenditure when compared to any other aspects of their PPE including the SCBA. More broadly, research supports that increases in boot mass have led to increases in energy costs [13,30]. These findings are echoed by the research of Pace et al. [8] who found the use of a minimalist style boot (lighter, shorter stack heigh, and minimal drop height from heel to forefoot) resulted in a significant decrease in VO_2_, respiratory exchange ratio, and perceived lower limb exertion while running as well as decreased ratings of perceived breathing exertion when walking and running. These findings are supported by previous research highlighting that loads carried on the feet generally incur the highest energy cost in both walking and running when compared to other modes of load carriage [81,82,83,84]. In an early study on the impact of boot mass on the energy costs of performing tasks, Soule and Goldman [81] observed increases in energy cost per kilogram of added boot mass that were up to four to six times those observed per kilogram of added body mass. These findings were supported by those of Holewijn, et al. [83] who reported oxygen costs per added kilogram of boot mass to be approximately two to five times greater than those associated with a kilogram of additional body mass. Chiou et al. [30] who, while likewise found a great energy cost imparted by heavier boots, found significant decreases in energy costs if more flexible soles were worn. As such, the research suggests that lighter boots with more flexible soles may reduce the energy costs of a given task.

However, these findings may be more common in boots. A study by Alferdaws et al. [28] comparing the physiological differences between three weighted shoes, did not find any significant differences in energy expenditure. Conversely, a similar study by Al-Ashaik et al. [27] using the same shoes in a similar population of university workers found significantly greater heart rates and ratings of perceived exertion in the heavier shoes. Both of these studies had relatively small sample sizes (n = 7 [27]; n = 10 [28]). As such, further research is required to elucidate the impacts of shoe (as opposed to boot) mass more clearly on the energy costs of performing physical tasks.

The mass of the occupational shoes may also contribute to a thermal effect. Given that the volume of evidence suggests heavier boots elicit greater energy costs, the thermal impacts of this increased work may contribute to heat gain. This supposition is supported by Al-ashaik et al. [27] who found higher aural-canal temperatures for participants who wore the heavier duty safety shoes. Subsequently, wearing boots has been found to have a thermal effect. Lee et al. [35] found sweat rates and body temperatures were significantly greater when firefighters wore their boots than other associated trial conditions (which included other PPE—including SCBA—with and without boots). Thus, the authors concluded that wearing boots led to greater heat production as the distance of the foot from the centred mass of the body makes heat distribution mechanically inefficient. These thermal effects are of importance when considering occupational environments. Soldiers, for example, can be deployed to varying climates, from cold alpine mountains to hot desert environments to humid tropical environments [22]. More locally, a police officer in one part of a country will face different daily temperatures than a police officer in another. Consider Australian police officers serving in the Northern Territory, where average annual temperatures range from 12 °C to 36 °C versus officers serving in Tasmania, where the mean annual temperatures range from −3 °C to 18 °C [85]. For the officers serving in Tasmania, increased heat production from the boots is less likely to be of concern and may even be of benefit.

Changes in muscle activity appear to not only be variable based on footwear mass, cushioning, sole and shaft, but also task and external temperature. In the three studies that compared the impacts of footwear on muscles of the lower back, the findings were variable. Huebner et al. [33] found a significant decrease in back musculature activity with increased shoe cushioning while conversely Svenningsen et al. [46] found a significant increase in back muscle activity when participants wore a rocker style (unstable shoe). When investigating the impacts of shoe mass, Al-Ashaik et al. [27] found no changes in back muscle activity (although the trapezius and biceps brachii muscle activity increased) during a specific lifting task. As such, if only looking at a specific muscle group, for example, the lower back which is a known site of injury in military [86], law enforcement [87], and fire and rescue personnel [18], cushioning in the footwear may decrease lower back muscle activity, wearing a rocker style shoe may increase activity, and, if performing a lifting task, the mass of the shoe may not impact on lower back muscle activation. Thus, the nature of the footwear (cushioning, stability, and mass) and the nature of the task (walking, lifting) produced varying results for a specific group of muscles. Of note, none of these three studies included boots. As such, the findings are expected to again vary. For example, with stiff boots found to reduce ankle joint range and increase hip joint range, when lifting an object, an increase in muscle activity may be required due to changes in joint range.

Of the two studies that included the quadriceps, hamstrings group, and gastrocnemius muscles, results were again variable as were footwear factors. For example, Dobson et al. [31] found that the type of sole and shaft utilised for a boot had significant impacts on the assessed muscles and their peak activity and onsets while walking [31]. Overall, the boots that were either all flexible or all stiff were likely to have higher muscle activity and earlier onsets of muscle engagement in preparation for initial foot-ground contact. Subsequently, in the study by Huebner et al. [33] which included walking with different shoe cushioning, muscle activity for the same muscles was not significantly different. Thus, a boot that has both flexible and stiff elements as a mixed construction (e.g., a stiff shaft with a flexible sole or a flexible shaft with a stiff sole) may be a better option than all flexible or all stiff [31] while differences in cushioning may not lead to changes in muscle activity.

The temperature in which tasks take place may also influence muscle activity patterns. In the study by Al-Ashaik et al. [27], biceps brachii muscle activity, for the same task wearing light to heavy shoes, differed following changes in climatic conditions (20 °C to 30 °C WBGT). Where the lighter shoes had the lower activity in the cooler condition (20 °C WBGT), the medium mass shoes had the lower activity (lower than light and heavy shoes) in the hotter condition (30 °C WBGT). This finding warrants consideration given that occupational footwear impacts are often only performed in one climate, while some populations, like firefighters, who are well represented in this review, will work in environments where the temperature reaches over 50 °C at 0.3 m above the floor [88].

Finally, muscle activity was found to vary when the subject was walking on gravel versus soft surfaces [31]. These findings are not surprising given research in military populations finding that load carriage task energy costs change depending on the type of terrain traversed. For example, leading work by Soule and Goldman [89], reviewing the energy costs for load carriage over sealed roads, dirt roads, light and heavy bush, swamp, and loose sand with loads of 8 kg, 20 kg, and 30 kg carried at speeds ranging from 2.4 km/h to 5.5 km/h, was used to create terrain coefficients based on energy costs. In increasing order of associated energy costs, the terrain coefficients were ranked as: sealed roads (1.0), dirt roads (1.1), light bush (1.2), heavy bush (1.5), swamp (1.8), and loose sand (2.1). As such, the terrain over which the individual wears their footwear will lead to different muscle activity and energy costs and thus bear consideration when types of occupational footwear are trialled.

While some studies [32] did use occupational tasks as an activity, the actual impacts of occupational footwear on actual occupational tasks themselves were limited. Smaller clogs resulted in quicker short distance run times of hospital staff to an emergency department elevator while lighter shoes were associated with a heavier average mass lifted and higher MAWL [27,28]. These limited results clearly highlight how little research has been done on the impact of occupational footwear on actual occupational task performance itself.

Overall, the volume of evidence from this review suggests that occupational footwear could have significant impacts on physical task performance both directly (e.g., running a short distance and lifting loads) and indirectly (e.g., reducing ROM, increasing GRF and energy and thermal costs). However, further research is needed to specifically examine occupational footwear impacts on actual occupational tasks given the findings of impacts of results based on thermal temperatures, terrain type, etc.

### 4.2. Injury Risk

Research associating occupational footwear with injuries is generally inconclusive. Studies in automotive workers found no relationships between lower limb injuries (plantar fasciitis, hip pain, and foot and ankle disorders) and firmness of heel, outer sole, or stiffness insole [53,54], although outer sole stiffness was associated with new onset foot and ankle disorders [54]. In nursing shoes, a thin sole was claimed to increase the number of discomfort complaints for the back and lower limb [62] while harder footwear was said to increase the risk of lower extremity self-reported fatigue [62]. Conversely, different types of boots worn did not impact injury risk [47,50,66] although boot type may influence the type of injuries reported. For example, gumboot wearers were more likely to have ball of foot pain, lateral malleolus pain, and arch pain while leather boot wearers were more likely on the other hand were significantly more likely to have navicular pain bunions sole pain heel pain and cuboid pain [47]. Considering this, with some injuries more prevalent than others in a given population (e.g., foot blisters being a common injury in military soldiers marching with load [90,91]) differences in injuries between shoe types may be impactful at an occupational level and may explain why a military study on injuries did find some differences in injuries between a leather boot and a hot weather boot [92].

Footwear comfort and fit were found to be associated with pain. In nurses low shoe comfort scores were associated with an increase in the reported presence of foot and ankle pain [29,51], knee pain [29] and hip/thigh pain [29]. Similarly, workers who rated their boot fit as very poor to reasonable were associated with an increased risk of hip pain while those who rate their boots as uncomfortable were associated with increased risks of foot pain [48]. Thus, relationships appear to exist between reported pain and footwear comfort and correct fit. Considering these findings, the use of rocker shoes (unstable shoes) in nurses has been found to reduce levels of pain and discomfort in nursing and hospital worker populations [55,58]. Conversely, boots with wool liners were found to be more likely to result in complaints of chaffing and higher discomfort when compared to boots without the liners [60].

When considering the mechanisms through which occupational footwear may contribute to injury risk, the research suggests that heavier footwear (in this case firefighter boots) may increase the risk of tripping over an obstacle [13,30,56]. These findings are of concern when considering that the occupational loads and general clothing carried and worn by firefighters can increase their risk of tripping over an object [93] The review did find means of mitigating the risk of slips, trips and falls in regards to occupational footwork and include the provision of slip resistant footwear (as opposed to recommendation to wear) [56] and the use of minimalist footwear which may reduce falls, and improve static and dynamic balance which may contribute to a fall [63]. Furthermore, minimalist shoes may decrease muscular exertion [63] which may in turn reduce muscle stressing injuries [18].

With regards to injury risk, the impacts of increases in stride length and GRFs, decreases (e.g., ankle) and resultant increases (e.g., knee and hip) in ROM, changes in balance strategies and requirements, impacts of physiological loads (e.g., HR and thermal), and changes in muscle activity may all contribute to increases in injury due to footwear worn. Therefore, if footwear is being developed for a specific occupation, understanding the most common natures and mechanisms of injury warrant consideration when determining the impacts of footwear on physical task performance and injury risk. Furthermore, second order effects that may impact physical task performance and mechanisms of injury (e.g., loss of ankle range leading to poor foot clearance and a subsequent injury reported as a trip) must be evaluated.

### 4.3. Limitations

One limitation of this review is the lack of research comparing occupational footwear to specific occupational task performances. While measures, such as energy expenditure and respiratory exchange ratios, etc., were measured and found to improve with the use of certain occupational footwear (in this case a minimalist style boot), it remains to be seen how this footwear affects actual physical task performance. Furthermore, of the 50 articles meeting the criteria for this review, only three articles investigated the impacts of footwear on occupational tasks specifically. Future research is necessary to compare occupational footwear impacts on actual physical task performances (e.g., victim drag or loaded march in a military context). Additionally, though it appears that occupational footwear may have a negative impact on variables that can contribute to occupational injury (e.g., GRF and ankle ROM), when analysing injury rates, few differences between footwear types are seen. The difficulty in elucidating injuries based on footwear type is noted by Cavanagh [94] who states that footwear effects on an injury can occur at a very subtle level.

## 5. Conclusions

The results of this scoping review suggest that occupational footwear can impact physical task performance and injury risk. However, the impacts largely depend on the type of footwear being worn, the conditions they are being worn in, and the outcome measures used. Consistent findings are a change in gait mechanics with occupational footwear, reduced ankle joint range of motion with a higher shaft height of stiffer materials and greater energy expenditure and poorer obstacle clearance with heavier footwear. These findings have the potential to increase the risk of injuries associated with slips and trips and muscle stress. Specifically, rubber boots tend to perform poorer than leather boots as do wader boots when compared to hiking boots and polyurethane boots when compared to styrene-butadiene boots. Occupationally, larger size clogs may negatively impact on the speed of running over a short distance while heavier shoes may reduce lifting capability. Further research, specifically focusing on the impacts of occupational footwear on actual occupational tasks is needed.

## Figures and Tables

**Figure 1 ijerph-19-10703-f001:**
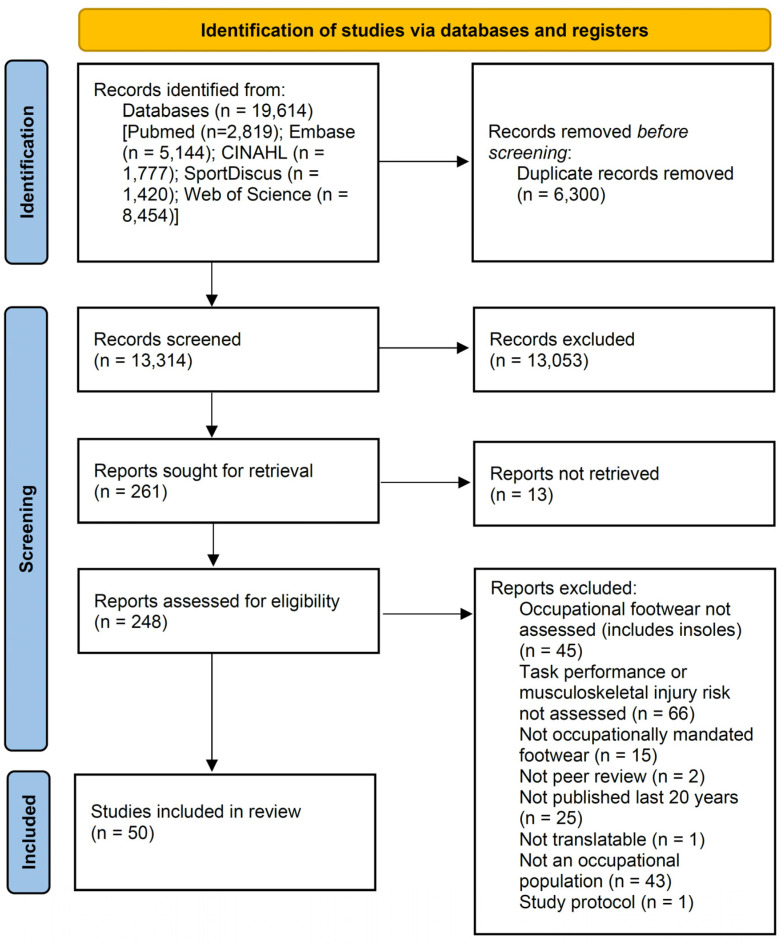
PRISMA Flow diagram showing results of the search, screening, and selection process [26].

**Table 1 ijerph-19-10703-t001:** An example of the search string used in PubMed.

Database	Search Terms
PubMed	“Boot*”[Title/Abstract] OR “Shoe*”[Title/Abstract] OR “Footwear”[Title/Abstract]) AND (“Occupation*”[Title/Abstract] OR “Profession”[Title/Abstract] OR “Trade*”[Title/Abstract] OR “Job”[Title/Abstract] OR “Work*”[Title/Abstract] OR “Safety”[Title/Abstract] OR “Nurses”[Mesh] OR “Miners”[Mesh] OR “Emergency Responders”[Mesh] OR “Military Personnel”[Mesh] OR “Farmers”[Mesh])

**Table 2 ijerph-19-10703-t002:** Inclusion and exclusion criteria.

Inclusion	Exclusion
Target population was a specific, paid, occupation such as nurse, construction worker, military personnel, etc.;*^1^ and Studies were published in English; and Studies were peer reviewed; and Studies contained information regarding footwear mandated by the occupation.	Studies did not specifically assess the physical impact of footwear on occupational task performance or other physical tests of human performance (e.g., vertical jump, range of motion); or The study did not assess the impact of occupational footwear on injury risk; or The study included the use of prescriptive prophylactic/ergonomic devices, such as insoles, are not deemed occupational footwear; or The study was of a published protocol; or The study primarily investigated risk of slips, trips, and falls with footwear potentially discussed as a causative factor; *^2^ or The study only reported on subjective findings/worker feedback on footwear.

*^1^ Athletes were not considered an occupation as their footwear was based on sporting requirements. *^2^ Relevant information from these articles were used to inform discussions as appropriate.

**Table 3 ijerph-19-10703-t003:** Characteristics and key findings of studies assessing occupational footwear and impact on task performance and musculoskeletal injury risk.

Author	Study Design	Boot Construction	Impact on Task Performance	Impact on Injury Risk
Al-Ashaik et al. [27] 2015	Quasi-experimental n = 7 University workers Age = 29.3 ± 3.9 yr Height = 166.1 ± 3.3 cm Mass = 70.7 ± 4.2 kg	3x different shoes all made by **Shelterall** Company, Italy: **Light duty (Reference):** similar to the regular leather shoes full leather with double density with padded collar rubber sole steel toecaps low cut pair mass: 0.9 kg **Medium duty:** genuine full leather with double density padded collar polyurethane moulded sole steel toecap low cut pair mass:1.05 kg **Heavy duty:** waxy full grain leather with double density with padded collar polyurethane moulded sole double steel toecaps high cut pair mass: 1.45 kg	Interaction between environmental temperature and type of safety boot had significant effect on Maximum acceptable weight of lift (MAWL) F(2,24) = 5.4, *p* < 0.012 MAWL while wearing heavy-duty shoes in 30 ° C was significantly less than wearing light-duty shoes at low temperatures *p* < 0.013 Aura Canal temperature was significantly higher in heavy duty shoes compared to medium (*p* = 0.02) or light (*p* < 0.0001) duty shoes % Maximum voluntary contraction (MVC) Significantly lower in biceps brachii while performing 1 lift/minute wearing light-duty safety shoes than wearing heavy-duty (*p* < 0.04) at 20 ° C Significantly lower in biceps brachii while performing 5 lifts/min wearing medium-duty safety shoes than when wearing heavy-duty (*p* < 0.00) at 20 ° C Significantly lower in biceps brachii while performing 1 lift/min wearing medium-duty shoes compared to heavy-duty (*p* 0.022) at 30 ° C Significantly lower in biceps brachii while wearing light-duty shoes compared to heavy-duty (*p* < 0.002) at 30 ° C Significantly lower in trapezius muscle group while wearing light-duty safety shoes compared to heavy-duty (*p* < 0.04) at 20 °C Rating of Perceived Exertion in 30 °C was significantly lower while wearing light-duty safety shoes compared to heavy-duty (*p* < 0.01)Safety shoe discomfort rating Light duty shoes rated as significantly more comfortable than medium-duty safety shoes (*p* < 0.0001) Light duty shoes rates as significantly more comfortable than heavy-duty safety shoes (*p* < 0.001) Medium-duty safety shoes were rated more comfortable during lifting than heavy-duty safety shoes (*p* < 0.0001) Mean weight lifted per shoe: **Light-Duty** 20 °C: 26.3 kg 30 °C: 25.2 kg Average: 25.1 kg **Medium-Duty** 20 °C: 25.1 kg 30 °C: 23.1 kg Both: 24.2 kg **Heavy-Duty** 20 °C: 23.4 kg 30 °C: 21.9 kg Both: 22.7 kg	
Alferdaws et al. [28] 2020	Quasi experimental n = 10 University workers Age = 29.7 ± 3.3 yr Height = 167.3 ± 7.1 cm Mass = 72.2 ± 7.2 kg	3x different shoes all made by **Shelterall** Company, Italy: **Light duty (Reference):** similar to the regular leather shoes” full leather with double density with padded collar rubber sole steel toecaps low cut pair mass: 0.9 kg **Medium duty:** genuine full leather with double density padded collar polyurethane moulded sole steel toecap low cut pair mass: 1.05 kg **Heavy duty:** waxy full grain leather with double density with padded collar polyurethane moulded sole double steel toecaps high cut pair mass: 1.45 kg	MAWL significantly higher while wearing light duty shoes compared to wearing heavy duty shoes (*p* < 0.041) No significant differences between medium and heavy or medium and light duty shoes Shoe type had no effect on respiration rate, minute ventilation, VCO_2_, relative VO_2_ or heart rate Shoe discomfort rating Heavy-duty shoes produced significantly more discomfort than medium or light duty shoes (*p* < 0.000 for both) Medium-duty shoes produced significantly more discomfort than light-duty shoes (*p* < 0.000)	
Anderson et al. [29] 2021	Operating theatre practitioners: n = 147 **Female** (n = 111) Height = 163.0 ± 9.0 cm Mass = 70.0 ± 14.7 kg BMI = 26.0 ± 6.2 kg/m^2^ **Male** (n = 36) Height = 176.0 ± 10.0 m Mass = 83.7 ± 14.5 kg BMI = 27.1 ± 3.5 kg/m^2^	Four main footwear types: Washable clog—usually made from EVA Standard clog—usually leather/microfibre upper Trainer Dress shoe/flat Other category—inclusive of Wellington boots and orthopaedic sandals		Greater footwear comfort corresponded to decreased risk of suffering from Hip/thigh pain (OR = 0.9, 95% CI 0.7–1.0) Knee pain (OR = 0.9, 95% CI 0.7–0.9) Foot pain (OR = 0.8, 95% CI 0.7–0.9)
Anderson et al. [62] 2017	Narrative Review	The study notes that There are clear limitations to the current studies There is a lack of methodological standardisation, particularly in studies looking at solutions (i.e., flooring and footwear) is contributing to the conflicting results between studies. This is due to both a lack of detail in the reporting of some methods and to the range of techniques used to measure the same dependent variables		Harder footwear increased the risk for lower extremity self-reported fatigue 2.6-fold (CI = 1.3 to 5.3) versus footwear with a low hardness level A thin sole in nursing shoes increased the number of discomfort complaints in the back, thigh, knee, and shin Width of footwear had an impact on the pressure distribution in the toes and that an arch support increased the area of the foot in contact with the shoe, reducing peak pressures Higher EMG values for the peroneus longus and gastrocnemius muscles were found in the footwear with the stiffest midsole
Armand et al. [55] 2014	Randomised Control Trial n = 40 (36 female, 4 male) **Intervention** Age = 44.5 ± 7.9 yr Height = 162.1 ± 9.1 cm Mass = 66.2 ± 11.3 kg BMI = 25.1 ± 3.9 kg/m^2^ **Control** Age = 46.8 ± 8.8 yr Height = 164.8 ± 7.8 cm Mass = 71.6 ± 13.7 kg BMI = 26.5 ± 5.5 kg/m^2^	**Intervention group:** Wore unstable shoes **Control group:** Wore conventional sports shoes (model Adidas Bigroar)		Intervention group: Significant decrease in pain while walking in lab barefoot (*p* = 0.037) Significant decrease in pain while walking in lab shoes (*p* = 0.001) Significant decrease in daily logbook of pain (*p* = 0.005) No significant decrease in pain during the last 24 h (*p* = 0.199) The Intervention showed a greater improvement in disability scores, but not statistically significantThe rate of satisfaction (satisfied and very satisfied) was 79% in the IG compared to 25% in the CG (*p* = 0.002).
Bell et al. [56] 2019	A two-arm cluster Randomised Controlled Study **Intervention** n = 6629 school district workers **Control** n = 4818 school district workers	Intervention consisted of providing slip resistance footwear rather than recommending workers to utilise slip resistance footwear and purchase them on their own.		Intervention significantly reduced probability of slipping injury (Oradj = 0.33, 95% CI 0.17–0.63).
Chander et al. [63] 2019	Narrative Review Discussed potential benefits of utilising minimalist boots	Heel-to-toe drop lower drop aids in neutral position of ankle and foot helping postural stability. Heel height: lower heel height aids in neutral position helping postural stability. Midsole: thin and firm midsole aids in better proprioception and somatosensory feedback. Insole: textured insole aids in better proprioception and somatosensory feedback. Foot-bed shape: heel seat lengths and heel wedge angle that promotes greater contact with the foot minimised foot pressure. Mass; lower mass aids in less energy expenditure and lower rate of muscular fatigue. Boot shaft: more flexible boot shafts that extend over the ankle can allow further joint ROM and promote joint position sense.	Minimalist boot performed better in minimising slip-induced falls, improved static, and dynamic balance, and lowered muscular exertion.	
Chiou et al. [30] 2012	Quasi-experimental n = 27 Firefighters 13 female, 14 male **Female** Age = 33.2 ± 4.4 yr Height = 166.6 ± 5.0 cm Mass = 67.9 ± 8.0 kg **Male** Age = 28.4 ± 5.5 yr Height = 178.5 ± 5.8 cm Mass = 94.6 ± 15.6 kg	Four models of firefighter boots conforming to NFPA standards for structural firefighting were selected for the study (NFPA, 2007). These boots were pull-up bunkers boots that were commercially available: Hybrid upper with a combination of leather and fabric upper and less flexible soles (HS): 2.05–2.48 kg Leather upper with less flexible soles (LS): 2.46–2.93 kg Leather upper and more flexible soles (LF): 2.56–3.10 kg Rubber upper with more flexible soles (RF): 3.36–3.82 kg	Of all 168 trials, 19 (11.3%) tripping incidents occurred. The following number of trips was found for each model: (a) HS = 6 (b) LS = 4 (c) LF = 6 (d) RF = 3 Results from the ANCOVA revealed a significant boot mass effect on trailing toe clearance for high (*p* < 0.02) and low obstacle heights (*p* < 0.003): Significantly shorter trailing toe clearance with heavier boots: For every 1 kg increase in mass there was an estimated 2.9 cm and 4.4 cm decrease in trailing to clearance for high and low obstacles respectively Significant interaction of boot mass on lead heel contact velocity (*p* < 0.02) and pre-obstacle distance (*p* < 0.05) while clearing a high obstacle: Participants placed their trailing foot closer to the obstacle before crossing while wearing heavier boots Heavier boots resulted in greater lead heel contact velocity Significant boot mass effects (*p* < 0.05) were observed for: Greater VE with heavier boot mass (significant for male only) Greater VO_2_ with heavier boot mass (male and female) Greater relative VO_2_ with heavier boot mass (male and female) Greater VCO_2_ with heavier boot mass (both males and females)	
Chorsiya et al. [67] 2018	Conference abstract n = 25 male subjects		Multiple ANOVA results showed the significant influence of shoe characteristics (toe cap, sole of shoe, mass of the shoe and ankle type) and their interaction on the centre of pressure displacement determinants.	
Choukou et al. [68] 2013	Conference abstract n = 10 workers Age =: 23.3 ± 6.7 yr, BMI =: 24.0 ± 2.0 kg/m^2^ Shoe size range: 43–44	Four conditions: Barefoot Safety shoes respecting conventional standards (l) Comfort safety shoes (OREGON) Unstable safety shoes (MBT)	There was no significant difference in gait frequency under the different conditions (*p* > 0.05) Gait duration is greater when barefoot than shod (F (3, 116) = 4.7, *p* < 0.05). Heel strike peak of force was higher with MBT than the other conditions (F (3, 116) = 4.4, *p* < 0.05). Foot flat peak force was higher when barefoot than shod (F (3, 116) = 4.2, *p* < 0.05) MBT was similar to other footwear conditions (*p* > 0.05). Toe off peak of force was higher for MBT (F (3, 116) = 11.4, *p* < 0.05)	
Choukou et al. [69] 2013	Conference abstract n = 10 workers Age =: 23.3 ± 6.0 yr Height: =: 1.8 ± 0.1 m Mass: =: 77.9 ± 8 kg, Shoe size range: 43–44	Four conditions: Barefoot Safety shoes respecting conventional standards (l) Comfort safety shoes (OREGON) Unstable safety shoes (MBT)	Anteroposterior magnitude, total area, length and velocity of centre of pressure were significantly higher when wearing MBT (F(3,116) = 10.5;94.3; 94.3; 9.5; respectively *p* < 0.05).	
Dobson et al. [13] 2017	Systematic Review 18 studies investigating the effect of boot design on walking	Comparison between multiple boots with focuses on following categories: Shaft Height Shaft Stiffness Boot Mass	**Shaft height** Shaft height could influence an individual’s foot and ankle ROM thereby altering lower limb mobility while walking Walking in pull-up bunker firefighting boots, compared to low-cut running shoes, significantly reduced ball of foot flexion-extension and ankle plantar flexion-dorsiflexion ROM Higher shafted firefighting boot led to increased ball of foot abduction-adduction and ankle inversion-eversion ROM in the frontal plane compared to when the participants wore the running shoe **Shaft stiffness** Manipulation of shaft stiffness in hiking boots, military boots and basketball boots has been found to significantly alter ankle ROM A more flexible shaft increased ankle ROM during walking and a stiffer shaft reduced it Restricting ankle joint motion is also thought to affect the hip by causing individuals to rely on hip motion changes to maintain balance A military boot with a softer, more flexible shaft that allowed more ankle ROM was shown to increase power generation during push-off at the ankle joint by 33% compared to when participants wore a military boot with a stiffer shaft **Boot mass** Boot mass is the most variable element of work boot design and can typically range between 1 and 4 kg Heavier footwear has been shown to alter the way individuals walk, particularly kinematic parameters characterising walking and oxygen consumption Increased heel contact velocities and reduced trailing limb toe clearances have been found Energy expenditure while walking increases with an increase in footwear mass	**Shaft height** Introducing a boot with a higher shaft, compared to a boot with a lower shaft, reduced the amount of ankle injuries incurred by Royal Marine recruits further supporting the notion of boot shaft height influencing ankle stability Wearing combat assault boots led to significantly higher peak pressures (kPa) being generated under metatarsals and higher peak loading rates under all metatarsal heads compared to wearing a gym trainer **Shaft stiffness** Enclosing the ankle and shank with a stiffer boot shaft can create a protective effect in the lateral direction minimises lateral ligament ankle sprains
Dobson et al. [59] 2018	Cohort n = 358 underground coal miners Age = 39.2 ± 9.6 yr Height = 178.7 ± 5.8 cm Mass = 92.8 ± 12.6 kg			Lower back was significantly related to Heel breadth (χ^2^ = 8.1, *p* = 0.016) Heel girth circumference (χ^2^ = 15.4, *p* = 0.038) Foot pain was significantly related to Ball of foot girth circumference (χ^2^ = 37.4, *p* = 0.021), specific to bunions Instep Height (χ^2^ = 8.33, *p* = 0.034), specific to calluses Hip pain was significantly related to Instep height (χ^2^ = 12.7, *p* = 0.019) Ankle pain occurrence was significantly related to toe angle (χ^2^ = 36.5, *p* = 0.013). Instep height, ball of foot girth circumference, foot breadth, and toe angle were significant predictors of low back pain, hip pain, and foot problems. However, the *R^2^* were low (0.062, 0.157, and 0.066 respectively).
Dobson et al. [47] 2017	Cross-Sectional n = 358 Underground coal miners (335 male, 3 female) Age =: 39.1 ± 10.7 yr Height = 178.0 ± 31.0 cm Mass = 92.1 ± 13.7 kg	Participants were divided into two groups for analysis based on whether they chose to wear the employer-provided gumboot (n = 219 men and 3 women) or the other mandatory boot option of the leather lace-up boot (n = 109 men).		No significant difference between boots for: Presence of lower back pain Presence of hip pain Presence of knee pain Presence of ankle pain Presence of foot pain There were significant differences between locations of foot pain.Gumboot wearers were more likely to have: Ball of foot pain (χ^2^ = 12.87, *p* = 0.002) Lateral malleolus pain (χ^2^ = 6.44, *p* = 0.040) Arch pain (χ^2^ = 6.72, *p* = 0.035) Leather boot wearers were more likely to have: Corns (χ^2^ = 6.78, *p* = 0.034) Navicular pain (χ2 = 7.09, *p* = 0.029) Bunions (χ^2^ = 6.72, *p* = 0.035) Sole pain (χ^2^ = 10.14, *p* = 0.006) Heel pain (χ^2^ = 7.18, *p* = 0.028) Cuboid pain (χ^2^ = 15.17, *p* = 0.001)
Dobson et al. [48] 2018	Cross-Sectional n = 358 underground coal miners (335 male, 3 female) Age = 39.1 ± 10.7 years Height = 178.0 ± 31.0 cm Mass = 92.1 ± 13.7 kg			Participants with hip pain were more likely to rate their work boot fit as very poor, poor, or reasonable (χ^2^ = 11.9, *p* < 0.05). Participants with foot pain were more likely to rate comfort as uncomfortable to indifferent (χ^2^ = 18.4, *p* < 0.001).
Dobson et al. [31] 2019	Quasi-experimental n = 20 workers who habitually wore steel caped safety boots, 11 underground coal miners, 9 trade workers Age = 36.0 ± 13.8 yr Height = 174.8 ± 6.3 cm Foot Length = 23.8 ± 0.6 cm Foot Width = 9.2 ± 0.4 cm	Four work boot conditions: **Flexible shaft and stiff sole** Mass: 0.94 kg Shafter Height: 29.5 cm Shaft Stiffness: 1.1 N Shaft material: Leather and nylon blend Sole Flexibility: 20.3° **Stiff shaft and stiff sole** Mass: 0.98 kg Shafter Height: 30 cm Shaft Stiffness: 1.7 N Shaft material: Leather Sole Flexibility: 20.3° **Stiff shaft and flexible sole** Mass: 0.98 kg Shafter Height: 30 cm Shaft Stiffness: 1.7 N Shaft material: Leather Sole Flexibility: 30.2° **Flexible shaft and flexible sole** Mass: 0.94 kg Shafter Height: 29.5 cm Shaft Stiffness: 1.1 N Shaft material: Leather and nylon blend Sole Flexibility: 30.2°	**Muscle burst onset relative to initial contact** Main effect boot shaft type (*p* < 0.001) Main effect surface condition (*p* < 0.001) Interaction between boot sole type and surface condition (*p* = 0.003) Interaction between boot shaft, boot sole, and surface condition (*p* = 0.002) **Effects on muscle burst onset relative to initial contact when on gravel surface** Main effect of boot shaft type (*p* < 0.001) Main effect of boot sole type (*p* = 0.032) Boot shaft and sole interaction (*p* = 0.032) **Effects on muscle burst onset relative to initial contact when on soft surface** Main effect of boot sole type (*p* < 0.001) Boot shaft and boot sole interaction (*p* = 0.044) **Thigh muscle onsets** Stiff shaft on gravel surface resulted in earlier vastus lateralis (*p* = 0.047) and semitendinosus (*p* = 0.003) onset relative to initial contact Flexible sole with flexible shaft led to earlier semitendinosus onset (*p* = 0.004) on gravel Flexible sole with flexible shaft led to earlier vastus lateralis onset (p) compared to stiff sole on soft surface **Shank muscle onsets** Stiff sole and stiff shaft resulted in earlier activation of peroneus longus (*p* = 0.023), and later activation of gastrocnemius (*p* = 0.005) compared to stiff sole and flexible shaft Flexible shaft with flexible sole resulted in later tibialis anterior onset (*p* = 0.023) relative to initial contact on gravel surface compared to a stiff shaft Stiff sole and stiff shaft led to earlier peroneus longus onset (*p* = 0.005) compared to a flexible sole on a soft surface **Peak muscle activity** Main effect boot sole type (*p* = 0.041) Main effect surface condition (*p* < 0.001) Interaction of boot shaft and boot sole type (*p* < 0.001) Interaction of boot shaft type and surface (*p* = 0.035) Interaction of boot sole type and surface (*p* = 0.002) On gravel main effect of boot sole type (*p* = 0.029) On gravel interaction of boot shaft and sole type (*p* < 0.001) On soft surface main effect of boot shaft type (*p* = 0.026) On soft surface main effect of boot sole type (*p* = 0.009) On soft surface interaction effect of boot shaft and sole type (*p* < 0.001) **Peak thigh muscle activity** Gravel surface with stiff shaft and sole led to increase semitendinosus (*p* = 0.041) activity compared to flexible shaft On gravel stiff sole with stiff shaft led to increased semitendinosus activity (*p* = 0.028) On soft surface a stiff shaft and sole lead to higher semitendinosus activity (*p* < 0.001) compared to flexible shaft On soft surface flexible sole and shaft led to higher semitendinosus activity (*p* < 0.001) compared to stiff sole **Peak shank muscle activity** On soft surface with flexible sole ad shaft led to higher peak medial gastrocnemius peak activity compared to stiff shaft **Muscle burst duration** No significant main effects of boot shaft or sole type on duration of lower limb muscle burst **Heel Contact Velocity** No significant main effects of boot shaft or sole type on heel contact velocity **Ankle alignment at initial contact** Main effect of boot shaft type (*p* = 0.022) Interaction between boot shaft and sole type (*p* = 0.033) Interaction between boot shaft and sole type and surface condition (*p* = 0.041) On gravel surface significant main effect of boot shaft type (*p* = 0.010) On gravel surface significant main effect of boot sole type (*p* = 0.027) On gravel significant interaction between boot shaft and sole type (*p* = 0.027) Boot with flexible sole and stiff shaft led to a greater eversion angle at initial contact compared to flexible shaft (*p* < 0.001) Boot with stiff shaft and flexible sole resulted in greater eversion angle (*p* = 0.002) compared to a stiff boot sole	
Dobson et al. [2] 2020	Quasi-experimental n = 20 workers who habitually wore steel caped safety boots, 11 underground coal miners, 9 trades workers Age = 36.0 ± 13.8 yr Height = 174.8 ± 6.3 cm Foot Length = 23.8 ± 0.6 cm Foot Width = 9.2 ± 0.4 cm	Four work boot conditions: **Flexible shaft and stiff sole** Mass: 0.94 kg Shafter Height: 29.5 cm Shaft Stiffness: 1.1 N Shaft material: Leather and nylon blend Sole Flexibility: 20.3° **Stiff shaft and stiff sole** Mass: 0.98 kg Shafter Height: 30 cm Shaft Stiffness: 1.7 N Shaft material: Leather Sole Flexibility: 20.3° **Stiff shaft and flexible sole** Mass: 0.98 kg Shafter Height: 30 cm Shaft Stiffness: 1.7 N Shaft material: Leather Sole Flexibility: 30.2° **Flexible shaft and flexible sole** Mass: 0.94 kg Shafter Height: 29.5 cm Shaft Stiffness: 1.1 N Shaft material: Leather and nylon blend Sole Flexibility: 30.2°	Significant main effect of boot shaft type on perceptions of foot (*p* = 0.025) and ankle (*p* = 0.48) ROM. Significant main effect of boot sole type on perceptions of ankle support (*p* = 0.020)—No significant differences could be found on a post-hoc analysis. Significant association (χ^2^ = 11.8, *p* = 0.008) between boot type and identification of best boot:Flexible shaft and still sole was preferred boot Stiff shaft and still sole least preferred No significant associations were identified between boot shaft or sole types and selection of best boot or worst boot.	Significant main effect of boot shaft type (*p* = 0.043), boot sole type (*p* = 0.002), and foot region (*p* < 0.001) on: Contact area Contact time Peak pressure Pressure-time integral variables Stiff boot shaft resulted in significantly: Greater contact area and contact time under medial heel (*p* < 0.001) Greater pressure-time integral and smaller contact area under medial midfoot (*p* = 0.015) Greater contact time and peak pressure under middle metatarsals (*p* = 0.016) Flexible sole resulted in significantly: Greater peak pressure and pressure-time integral under medial heel (*p* < 0.001) Reduced pressure-time integral under the hallux (*p* = 0.004) Boot shaft and sole interactions significantly impacted: Pressures under lateral midfoot (*p* < 0.001 Pressures under medial metatarsals (*p* = 0.038) Pressures under lateral metatarsals (*p* = 0.009) Pressures under lesser toes (*p* < 0.001) A stiff sole with flexible shaft, when compared to stiff shaft resulted in significantly: Increased contact area under lateral midfoot Decreased peak pressure and pressure-time integral under medial metatarsals A stiff shaft with flexible sole when compared to a flexible shaft: Increased contact time under lateral midfoot A stiff shaft with flexible sole compared to flexible shaft and stiff sole had: Decreased peak pressure under lateral midfoot and metatarsals Increased peak pressure under lesser toes Flexible boot sole with stiff shaft compared to stiff sole. Led to increased contact area and peak pressure under lateral midfoot compared to stiff sole Flexible boot sole compared to a stiff boot sole led to a greater peak pressure under medial metatarsals.
Elbers et al. [57] 2020	Randomised Control Trial n = 50 healthcare professionals (21 male, 29 female). Randomised to different clog sizes **Size 38 clogs** 10 Male, 15 female Age = 36.0 ± 12.0 yr Height = 176.0 ± 10.0 cm **Size 47 clogs** Male 11, Female 14 Age = 38.0 ± 12.0 yr Height = 176.0 ± 10.0 cm	Randomised to either size 38 clogs or size 47 clogs.	Size 38 clogs completed simulated course to emergency department in 34.2 ± 4.9 s Size 47 clogs completed course in 38.8 ± 6.4 sec Mean difference of −4.4 (95% CI −7.1- −1.6) s. No further modifications when accounting for gender, age, height, own shoe size, fitness, or staff function.	No significant difference in comfort or adverse effects.
Garner et al. [32] 2013	Quasi-experimental n = 12 professional firefighters Age = 33.4 ± 6.8 yr Height = 179.0 ± 6.5 cm Mass = 95.8 ± 21.5 kg	Comparison between rubber and leather boots. Rubber mean mass per pair: 2.90 ± 0.20 kg. Leather mean mass per pair: 2.40 ± 0.20 kg.	Significant differences were found between boot types (F(1,11) = 3.522, *p* = 0.03). Rubber boots resulted in greater sway (anteroposterior and medial-lateral) parameters, increased decrement in peak torque in lower limbs (which could lead to increase in localised fatigue).	
Gell et al. [49] 2011	Cross-sectional n = 407 automotive workers (309 male, 98 female) Age = 48.4 ± 10.3 yr BMI = 29.4 ± 5.3 kg/m^2^			Significant differences in footwear between workers who reported feeling lower extremity fatigue at the end of the day and those that did not. Individuals with harder outsoles were more likely to report lower limb fatigue (*p* < 0.01) Having high hardness compared to low hardness increased odds of lower limb fatigue (OR = 2.6, 95% CI 1.3–5.3, *p* = 0.01).
Huebener et al. [33] 2014	Quasi-experimental n = 10 canteen workers Age Range = 25–48 yr, median = 38.5 yr	**Control Shoe:** Standard safety shoe **Test Shoe:** housed an exchangeable cushioning element in the heel of inner shoe sole. Cushioning element prescribed depending on mass. Four categories: <57 kg 58–79 kg 80–91 kg >91 kg Cushioning elements were then divided into three groups: Dummy insert Optimal prescription based on above categories Too soft—cushion provided from lighter mass class than subject’s actual mass	Cumulative muscle activity per distance travelled (CAMPD) (an indirect measure of required energy expenditure) was measured. Significant differences across shoes were found only in back muscles at preferred (F_(3,7)_ = 7.016, *p* = 0.016) and fast (F_(3,7)_ = 4.568, *p* = 0.045) with optimal damping showing the lowest values. While not significant too soft showed the lowest values in the leg muscles, while optimal and no damping were the most economical in the abdominal muscle groups. Control shoes showed the highest values across all conditions. Significant differences were found for normalised mean range for leg muscles at preferred (F_(3,7)_ = 8.256, *p* = 0.011) and fast walking velocities (F_(3,7)_ = 7.105, *p* = 0.016) with optimal dampening resulting in higher scores. Though not significant lowest values occurred in control and non-dampened shoes in all muscle groups, with exception of abdominal muscles with optimal dampening resulting in highest values during preferred walking speed, control shoes showing values during fast walking speed, and marginal differences at slow speed. Optimal and too soft damping led to reduced amplitude heel strike levels, but these were not significant. Test shoes tended to have an earlier onset of back muscle activity. Optimal cushioning suggested significantly and consistently smaller amplitude peaks at the back muscles (10/12 *t*-tests showing significant differences, mean effect of 1.00, individual results not disclosed).	
Irmańska [60] 2015	Cohort n = 45 firefighters **Group A** (n = 15) Age = 33.4 ± 3.5 yr BMI = 25.4 ± 2.7 kg/m^2^ **Group B** (n = 15) Age = 32 ± 5.5 yr BMI = 25.7 ± 2.9 kg/m^2^ Group C (n = 15) Age = 31.5 ± 2.48 yr BMI = 25.0 ± 2.9 kg/m^2^	**Group A Boot:** Novel liner with ventilation Polyester and Lyocell blend Upper mass range: 170–200 g/m^2^ Insole mass range: 170–400 g/m^2^ **Group B Boot:** Novel liner with no ventilation Polyester and Lyocell blend Upper mass range: 170–200 g/m^2^ Lower mass range: 80–400 g/m^2^ **Group C Boot:** Standard liner Wool Upper mass 400 g/m^2^ Lower mass 400 g/m^2^	No significant differences were identified for foot mobility between boots. Thermal sensations (*p* < 0.05) and moisture sensations (*p* < 0.01) were more strongly reported in Group C and Group B compared to Group A. Thermal sensations and moisture sensations were comparable between Group C and Group B. No significant differences between groups with Walking difficulty (χ^2^ = 7.71, *p* > 0.05) Kneeling/crouching difficulty (χ^2^ = 1.09, *p* > 0.05)	Significant difference between footwear with descriptions of chaffing (χ^2^ = 6.14, *p* < 0.05) with Group A noted as less chaffing Group C had significantly less comfort (presence of rough, sharp, or hard areas that could cause injury or irritation) than Group A or B (χ^2^ = 10.49, *p* < 0.05).
Irmańska and Tokarski [34] 2016	Quasi-experimental n = 40 males: 20 younger (Y) and 20 older (O) **Group Y:** Age Range = 20–30 yr Height Range =167–186 cm Mass Range = 66–90 kg Occupations: firefighters, drivers, and farmers. **Group O:** Age Range = 60–65 yr Height Range = 166–188 cm Mass Range = 60–95 kg Occupations: Farmers and security personnel.	Two boot types: **Type A: low cut, sandal-like protective footwear** Slip resistance: 0.31 Impact absorption by materials at ankle: not tested due to construction Sole tread: 3.31 mm Energy absorption at the heel: 36 = 37 J **Type B: Protective ankle boots** Slip resistance: 0.35 Impact absorption at ankle: 9.8 kN Sole tread: 3.36 mm Energy absorption at heel: 35–36J	No significant difference in hip ROM between Y & O groups while in footwear. Significant difference in knee ROM between Y& O. **Footwear A** Treadmill test L *t* = −3.86, *p* < 0.001, O > Y Stair climbing L *t* = −2.64, *p* < 0.05, O > Y Treadmill test R *t* = −3.66, *p* < 0.05, O > Y **Footwear B** Treadmill test L *t* = −4.67, *p* < 0.001, O > Y Stair climbing L *t* = −3.79, *p* < 0.001, O > Y Significant difference in talocrural ROM between Y & O. **Footwear A** Treadmill test L *t* = 4.11, *p* <0.001 Y > O Treadmill test R *t* = 2.08, *p* < 0.05 Y > O Stairclimbing R *t* = 2.01, *p* < 0.05. Y > O **Footwear B** Treadmill R *t* = 4.09, *p* < 0.001 Y > O Stairclimbing R *t* = 5.34, *p* < 0.001 **Compared to barefoot** Footwear A resulted in significantly more flexion at left and right hip joints during treadmill walking for group Y (*p* < 0.05) Footwear A resulted in significantly less L talocrural joint range during treadmill walking and stair climbing for group O (*p* < 0.05) Footwear B resulted in significantly less joint L knee joint range during treadmill walking for group Y (*p* < 0.05) Footwear B resulted in significantly less L talocrural joint range during stair climbing for group Y (*p* < 0.05) Footwear B resulted in significantly less R talocrural joint range during stair climbing for group O (*p* < 0.05) **Compared to Footwear A** Footwear B resulted in significantly more L talocrural joint range during stair climbing for group Y (*p* < 0.05)	
Knapik et al. [65] 2015	Systematic Review with meta-analysis. Comparison of military physical training before and after 1982 when running shoes replaced military boots as footwear during physical training.	Identified 12 data collection periods, three during the “boot” period of training and 9 post.		Identified two separate injury definitions (overall and lower extremity injuries). Meta analysis showed: Injury incidence in males of 26.2% (95% CI 23.1–29.3) training in boots Injury incidence in females of 54.0% (95% CI 48.9–59.1) training in boots Overall injury incidence in males of 27.1% (95% CI 22.1–32.7) training in shoes Overall injury incidence in females of 54.9% (95% CI 46.8–62.8) χ^2^ results showed Any injury male (RR shoe/ boot 1.04 (95% CI 0.91–1.18; *p* = 0.50) Lower extremity injury male (RR shoe/boot 0.91 (95% CI 0.64–1.30) *p* = 0.66) Any injury female (RR shoe/boot 0.94 (95% CI 0.85–1.05) *p* = 0.27) Lower extremity injury female (RR shoe/boot 1.06 (0.89–1.27) *p* = 0.51)
Knapik et al. [64] 2015	Narrative Review			Reviews injuries and running shoes in military populations over the transition from standard issue boots to running shoes in the U.S. Military in 1982. Cites previous literature showing that injury incidence was not significantly reduced with introduction of running shoes into PT: Any injury male (26% injury incidence before 1982, 27% after, *p* = 0.50) Lower extremity injury male (23% injury incidence before 1982, 21% after *p* = 0.66) Any injury female (54% injury incidence before 1982, 51% after, *p* = 0.27) Lower extremity injury female (42% before 1982, 45% after, *p* = 0.51)
Kocher et al. [36] 2020	Quasi-experimental n = 10 workers (8 male, 2 female) Age = 28.6 ± 6.0 yr Mass = 86.9 ± 19.0 kg Height = 182.0 ± 8.0 cm	4 boot styles: Hiker style (laces, shorter shank) with steel toe (HS) Hiker style with metatarsal guard (HM) Wader-style (slip on, taller shank) with steel toe (WS) Wader-style with metatarsal guard (WM) Wader style shanks were slightly rolled down to allow placement of gait analysis markers.	Boot style significant interaction with: Foot velocity (**Ascending** R: *p* = 0.03, effect size = 0.278) Hip ROM: (**Descending** L: *p* = 0.005, effect size = 0.373, R *p* = 0.01, effect = 0.331) Knee ROM: (**Descending** L: *p* = 0.004, effect size = 0.777, R *p* = 0.005, effect = 0.744) Ankle ROM: (**Ascending** L: *p* < 0.001, effect size = 0.777, R: *p* < 0.001, effect size = 0.681; **Descending** L: *p* < 0.001, effect size = 0.703, R *p* < 0.001, effect = 0.512 Differences between boot type: **WS** significantly less ROM than **HS** (ascending L and R ankle; descending R ankle), **HM** (ascending L and R ankle; descending L ankle) **WM** significantly less ROM than **HS** (ascending L and R ankle; descending L and R ankle; descending L and R knee), **HM** (ascending L and R ankle; descending L ankle; descending L and R knee), **WS** (descending left hip) Incline level and Boot significant interaction with: Stride length (**Ascending** L: *p* = 0.02, effect size = 0.23)	
Lee et al. [35] 2014	Quasi-experimental n = 8 firefighters Age = 39.4 ± 5.6 yr Mass = 74.2 ± 10.0 kg Height = 173.9 ± 3.8 cm VO_2Max_ = 42.0 ± 5.1 mL/kg/min Experience = 10.4 ± 7.0 yr	Tested various components of firefighter PPE with comparisons between firefighter boots and thin sandals.	Total sweat rate (*p* < 0.05), rectal temperature (*p* < 0.05), skin temperature (*p* < 0.05), heart rate (*p* < 0.05), and oxygen consumption (*p* < 0.05) varied significantly (both during exercise and recovery) across PPE worn The PPE condition of no boots (but other PPE worn such as self-breathing apparatus, helmet, and gloves) resulted in similar physiologic scores as no thermal clothing and no equipment and greater benefits compared to removing the breathing apparatus, helmet, or gloves	
Majumdar et al. [37] 2006	Quasi-experimental n = 8 infantry soldiers Age = 26.7 ± 2.7 yr Mass = 59.3 ± 5.1 kg Height = 164.8 ± 4.4 cm	Comparison between barefoot and standard issue military boots. Testing conditions also included wearing combat vest.	Military boot resulted in Longer step & length [cm] (R 67.3 vs. 62.9, *p* < 0.001; L 65.7 vs. 61.5, *p* < 0.001 & R 132.9 vs. 124.6, *p* < 0.001; L 132.8 vs. 124.2, *p* < 0.001, respectively) Slower cadence [steps/min] (R 100 vs. 105.4, *p* < 0.01; L 99.9 vs. 105.4 *p* < 0.01) Less total support time [%] (R 59.0 vs. 59.6, *p* < 0.05; L58.5 vs. 59.7, *p* < 0.01) Longer swing phase [%] (R 40.9 vs. 40.4, *p* < 0.05; L 41.5 vs. 40.3 *p* < 0.01 Less initial double support time [%] (R 8.8 vs. 9.4, *p* < 0.05, L 8.6 vs. 9.8 *p* < 0.05) More single support time [%] (R 41.5 vs. 40.3, *p* < 0.01) No significant difference between L single support time, step width, or forward velocity	
Muniz et al. [38] 2021	Quasi-experimental n = 24 male soldiers Age = 18.9 ± 0.6 yr Mass = 67.3 ± 8.6 kg Height = 170.0 ± 10.0 cm	Comparison between military boots made with styrene-butadiene rubber (SBR), polyurethane (PU). Compared in both unloaded and loaded (15 kg) conditions. Boot Characteristics **SBR** Mass: 561.49 g Energy absorption: 23 joules Hardness: 63 Density: 1.132 g/cm^3^ Midsole rear-foot thickness: 36 mm EVA insole thickness: 3 mm **PU** Mass: 380.33 g Energy: absorption 31 joules Hardness: 48 Density: 0.563 g/cm^3^ Midsole rear-foot thickness: 32 mm EVA insole thickness: 3 mm		Instantaneous loading rate (%BW/s): SBR Unload 24.5 ± 6.6; Load 25.1 ± 5.2 PU Unload 26.1 ± 4.3; Load 27.8 ± 4.1 *p* = 0.04, η^2^ = 0.04 Median power frequency (Hz): SBR Unload 21.8 ± 3.3; Load 22.4 ± 4.7 PU Unload 24.6 ± 2.8; Load 24.9 ± 2.2 *p* < 0.01, η^2^ = 0.16 No significant difference in first peak force (%BW) or significant interactions between load and footwear.
Muniz and Bini [39] 2017	Quasi-experimental n = 20 Army recruits Age = 18.9 ± 0.6 yr Mass = 67.3 ± 8.6 kg Height = 170.0 ± 10.0 cm	Compared three boot conditions: **Boot 1** SBR Mass: 631.8 g Rear-foot thickness: 30 mm EVA: 3 mm thickness **Boot 2** SBR Mass: 530.3 g Rear-foot thickness: 20.6 mm EVA: 3 mm thickness **Boot 3** PU Mass: 423 g Rear-foot thickness: 31.7 mm EVA: 3 mm thickness		Significant difference in one component of the vertical principal component analysis (Boot 1 = −0.095 ± 0.13; Boot 2 = −0.030 ± 0.15; Boot 3 = − 0.064 ± 0.11; *p* < 0.001). Significant difference in one component of the anteroposterior component analysis (Boot 1 = 0.05 ± 0.09; Boot 2 = −0.04 ± 0.10; Boot 3 = −0.03 ± 0.37). Significant difference in two components of the mediolateral component analysis between Boot 1 (PC2 −0.03 ± 0.05; PC4 −0.02 ± 0.04) and Boot 2 (PC2 0.04 ± 0.05; PC4 0.02 ± 0.04). Significant difference in comfort between Boot 1 (5.5 ± 1.7) and Boot 3 (7.7 ± 2.3) as measured by visual analogue scale.
Neugebauer and Lafiandra [40] 2018	Quasi-experimental n = 15 male soldiers	Comparison between military boots and athletic footwear across 4 loads—0 kg, 14 kg, 27 kg, and 46 kg.		Hardness of footwear did was not a significant predictor of max ground reaction force (*p* = 0.70). Type of footwear was a significant model factor, but did not improve predictions considerably (*r*^2^ = 0.892) over predictive. Model without footwear type (*r*^2^ = 0.891). The average absolute percent difference with and without footwear term were similar (4.8% and 4.7% respectively).
Oschman et al. [9] 2016	Quasi-experimental n = 20 male automotive workers Age = 33.2 ± 10.5 yr Mass = 80.1 ± 7.8 kg Height = 177.9 ± 3.9 cm Median foot size (range) 27.8 cm (26 cm–28.7 cm)	Tested three safety shoes: **Shoe 1** Steel safety cap Mass: 530 g No varying widths Little cushioning No insole Treadsole—PU No ergonomic specifics **Shoe 2** Aluminium safety cap Mass: 630 g Different widths Cushioning in heel and forefoot Insole present Treadsole—thermoplastic polyurethane (TPU) Mass dependent heel absorption **Shoe 3** Steel safety cap Mass: 720 g No varying widths Heel cushioning Insole present Combination of PU and TPU Rocker-bottom sole (curve in anterior-posterior direction)	Significant differences between trunk inclination 50th percentile between Shoe 1 (8.9 ± 2.2°) and Shoe 2 (6.7 ± 3.5°), *p* = 0.005, Shoe 1 and Shoe 3 (5.9 ± 2.4°), *p* < 0.001. Significant difference in the 50th percentile of hip flexion between all shoes (Shoe 1: 14.0 ± 3.6°, Shoe 2: 11.5 ± 3.9°, Shoe 3: 10.2 ± 2.8°) Shoe 1 and Shoe 2 *p* = 0.015 Shoe 1 and Shoe 3 *p* = 0.001 Shoe 2 and Shoe 3 *p* = 0.046 Significant differences in knee flexion (95th–5th percentiles) between Shoe 1 (62.3 ± 3.4°) and Shoe 2 (64.0 ± 3.6°), *p* = 0.008 and Shoe 2 and Shoe 3 (62.0 ± 4.3°), *p* < 0.001. No significant differences in 50th percentile knee flexion hip flexion ROM (95th–5th percentile) or trunk ROM (95th–5th percentile).	Significant differences between Shoe 1 and Shoe 2 in maximum plantar pressure (N/cm^2^): Rearfoot zone 1: 27.9 ± 3.1 vs. 24.2 ± 2.0 *p* < 0.001 Rearfoot zone 2: 19.7 ± 3.1 vs. 14.4 ± 2.6 *p* < 0.001 Midfoot zone 3: 4.7 ± 1.3 vs. 5.5 ± 1.0 *p* = 0.002 Midfoot zone 4 2.8 ± 0.7 vs. 4.5 ± 1.2 *p* < 0.001 Midfoot zone 5 2.9 ± 0.9 vs. 4.7 ± 1.5 *p* < 0.001 Forefoot zone 6 12.0 ± 5.7 vs. 17.9 ± 5.9 *p* < 0.001 Forefoot zone 7 25.0 ± 4.0 vs. 22.9 ± 3.5 *p* = 0.003 Significant differences between Shoe 1 and Shoe 3 in maximum plantar pressure: (N/cm^2^): Rearfoot zone 1: 27.9 ± 3.1 vs. 24.2 ± 2.9 *p* < 0.001 Midfoot zone 3: 4.7 ± 1.3 vs. 5.6 ± 1.1 *p* = 0.005 Midfoot zone 4: 2.8 ± 0.7 vs. 5.2 ± 1.5 *p* < 0.001 Midfoot zone 5: 2.9 ± 0.9 vs. 4.0 ± 1.1 *p* < 0.001 Forefoot zone 7: 25.0 ± 4.0 vs. 20.9 ± 3.5 *p* < 0.001 Significant differences between Shoe 2 and Shoe 3 in maximum plantar pressure (N/cm^2^): Rearfoot zone 2: 14.4 ± 2.6 vs. 18.1 ± 2.7 *p* < 0.001 Midfoot zone 4: 4.5 ± 1.2 vs. 5.2 ± 1.5 *p* = 0.002 Midfoot zone 5: 4.7 ± 1.5 vs. 4.0 ± 1.1 *p* = 0.002 Forefoot zone 6: 17.9 ± 5.9 vs. 14.0 ± 5.3 *p* < 0.001 Forefoot zone 7: 22.9 ± 3.4 vs. 20.9 ± 3.4 *p* < 0.001 Forefoot zone 8: 17.1 ± 6.5 vs. 19.8 ± 4.9 *p* < 0.001 Significant differences in centre of pressure posterior-anterior 95–5th percentile for all 3 shoes (Shoe 1 159.5 ± 10.8 mm, Shoe 2 149.1 ± 10.3 mm, Shoe 3 143.7 ± 10.5 mm): Shoe 1 and Shoe 2 *p* < 0.001 Shoe 1 and Shoe 3 *p* < 0.001 Shoe 3 and Shoe 3 *p* = 0.003 Significant difference in 50th percentile centre of pressure medial lateral: Shoe 1 (2.0 ± 1.7 mm) and Shoe 2 (3.5 ± 2.0 mm) *p* < 0.001 Shoe 2 (3.5 ± 2.0 mm) and Shoe 3 (2.0 ± 2.0 mm) *p* = 0.002 Significant difference in 95th–5th percentile centre of pressure medial lateral: Shoe 1 (22.1 ± 5.1 mm) and Shoe 3 (20.2 ± 4.7 mm) *p* = 0.022 Shoe 2 (22.2 ± 4.7 mm) and Shoe 3 (20.2 ± 4.7 mm) *p* = 0.001
Oliver et al. [41] 2011	Quasi-experimental n = 16 Reserve Officer Training Corps cadets (13 male, 3 female) Age = 21.0 ± 3.0 yr Mass = 79.0 ± 12.0 kg Height = 172.0 ± 10.0 cm	Comparison between bare feet, tennis shoes, and issued military boots.		No significant differences in the degree of knee valgus between conditions. Significant differences for ground reaction force as percentage of bodyweight (bare feet: 1646 ± 359%, tennis shoe: 1880 ± 379%, boot: 1833 ± 438%, *p* < 0.05).
Pace et al. [8] 2020	Quasi-experimental n = 14 male Reserve Officer Training Corps cadets Age Range 20–30 yr Mass = 86.2 ± 10.4 kg Height = 177.0 ± 6.0 cm Body Fat = 8.1 ± 3.2% Load VO_2Max_ = 46.6 ± 7.3 mL/kg/min No load VO_2Max_ = 47.1 ± 5.7 mL/kg/min	Comparison between minimalist style (MIN) and standard issue military boots.	Significant difference in respiratory exchange ratio (RER) between MIN (0.94 ± 0.06) and standard issue (1.00 ± 0.07) *p* < 0.01, Cohen’s *d* = 0.90. Significant difference in VO_2_ while running MIN 34.4 ± 3.3 mL/kg/min, standard issue 35.5 ± 3.5 mL/kg/min *p* < 0.05, Cohen’s *d* = 0.31 Significant different in rating of perceived exertion (RPE) in breathing during: walking: MIN 2.0 ± 0.9, standard issue 2.4 ± 1.2, *p* < 0.05, Cohen’s *d* 0.33 running: MIN 4.0 ± 1.8, standard issue 4.8 ± 1.5, *p* < 0.01, Cohen’s *d* = 0.43 Significant difference between RPE in legs while running MIN 4.4, standard issue 5.7 ± 1.7 *p* < 0.05, Cohen’s *d* = 0.69.	
Park et al. [10] 2015	Quasi-experimental n = 12 (8 male, 4 female) firefighters **Male:** Age = 28.6 ± 8.3 yr Mass = 85.5 ± 15.7 kg Height = 183.5 ± 3.8 cm **Female:** Age = 31.5 ± 13.5 yr Mass = 68.3 ± 14.3 kg Height = 170.8 ± 7.6 cm	Comparison between running shoes, rubber firefighting boots, and leather firefighting boots. **Running shoes** Mass: 0.71 ± 0.24 kg Flex resistance: 5.97 ± 2.28 N **Rubber boots** Mass: 3.15 ± 0.29 kg Flex resistance: 27.2 ± 1.2 N Collar height: 38 cm Outsole height: 3 cm at heel, 1.5 cm at forefoot Rubber in upper and outsole Metal toe cap and metal shank **Leather boots** Mass: 3.02 ± 0.19 kg Flex resistance: 34.4 ± 2.6 N Collar height: 34 cm Outsole height: 3.5 cm at heel, 1.8 cm at forefoot 60%kevlar/40% Nomex fabric for upper Metal toe cap and metal shank	Significant differences existed between running shoes and both rubber and leather boots in: Normalised anterior-posterior excursion (67.24% running shoes, 64.27% rubber, 63.35% leather) *p* < 0.001 Centre of pressure velocity (58.55 cm/s running shoes, 55.07 cm/s rubber, 53.42 cm/s) *p* < 0.001 Stride time (1.11 sec running shoes, 1.17 sec rubber, 1.17 sec leather) *p* < 0.001 Stance phase (59.92% running shoes, 61.31% leather) *p* < 0.001. No significant difference between running shoes and rubber (59.74%) Double support (8.96% running, 10.5% leather) *p* < 0.001. No significant difference between running shoes and rubber (8.9%) Significant differences existed between rubber and leather boots in: Stance phase (59.74% rubber, 61.31% leather) *p* < 0.001 Double support (8.9% rubber, 10.5% leather) *p* < 0.001	
Park et al. [43] 2015	Quasi-experimental n = 12 (8 male, 4 female) firefighters **Male:** Age = 28.6 ± 8.3 years Mass = 85.5 ± 15.7 kg Height = 183.5 ± 3.8 cm **Female:** Age = 31.5 ± 13.5 years Mass = 68.3 ± 14.3 kg Height = 170.8 ± 7.6 cm	Comparison between running shoes, rubber firefighting boots, and leather firefighting boots. **Running shoes** Mass: 0.71 ± 0.24 kg Flex resistance: 5.97 ± 2.28 N **Rubber boots** Mass: 3.15 ± 0.29 kg Flex resistance: 27.2 ± 1.2 N Collar height: 38 cm Outsole height: 3 cm at heel, 1.5 cm at forefoot Rubber in upper and outsole Metal toe cap and metal shank	Range of motion in the sagittal plane significantly differed in: Hip (Running shoes 49 ± 1.21°, boots 50.99 ± 1.21°) *p* < 0.001. Indicates more hip flexion in boots Knee (Running shoes 64.14 ± 1.13°, boots 67.82 ± 1.14°) *p* < 0.001. Indicates more knee flexion in boots Ankle (Running shoes 42.25 ± 1.59°, boots 36.59 ± 1.60°) *p* < 0.001. Indicates less dorsiflexion in boots Ball of foot (Running shoes 63.71 ± 1.81°, boots 52.91 ± 1.86°) *p* < 0.001. Indicates less hallux flexion in boots Range of motion in the frontal plane significantly differed in: Ankle (Running shoes 16.19 ± 1.46°, boots 18.53 ± 1.49°) *p* = 0.026. Indicates more inversion in boots Ball of foot (Running shoes 18.30 ± 1.62°, boots 26.19 ± 1.67°) *p* < 0.001. Indicates more adduction in boots Range of motion in the transversal plane significantly differed in: Ankle (running shoes 22.05 ± 2.05°, boots 17.97 ± 2.07°) *p* < 0.001. Indicates more extra-rotation in boots Ball of foot *p* (running shoes 8.15 ± 0.904°, boots 11.10°) < 0.001, Indicates more extra-rotation in boots	
Park et al. [42] 2019	Quasi-experimental n = 14 firefighters (11 male, 3 female) Age = 32.7 ± 12.3 yr Mass = 79.2 ± 13.4 kg Height = 177.3 ± 4.7 cm	3 leather boot heights were tested: Low: 25.4 cm Medium: 30.48 cm High: 35.56 cm	Greater ROM for low boots than high boots regardless of knee heights for hip, knee, and ankle (mean differences = 2.0 −5.3°; *F* = 5.398–5.648 *p* = 0.004–0.005). Greater knee ROM for low boots compared to higher during duckwalking (*F* = 6.67, *p* = 0.002) Greater ROM in low compared to high boots accounting for knee height in hip ROM during duckwalking (mean difference: 10.5–12.8°; *F* = 15.127, *p* = 0.006). Firefighters with taller knee height had significantly smaller ankle ROM in high boots (15.35°) than in low boots (17.53°) (*p* = 0.025) during ladder ascension.	
Schulze et al. [44] 2014	Quasi-experimental n = 32 soldiers Age Mean (Median) = 29.0 (26.0) yr Mass Mean (Median) = 81.6 (81.0) kg Height Mean (Median) = 177.8 (179.0) cm	5 shoe variations compared: **Dress shoe** Mass: 530 g Cow leather Rubber sole 3-hole lacing **Combat boot** Mass: 1135 g Adherent rubber sole Leather with leather lining Bolstered boot leg 8-hole lacing **Outdoor athletic shoe (old design)** Mass: 500 g Leather Nubby rubber sole Ankle padding 6-hole lacing **Outdoor athletic shoe (new design)** Mass: 720 g Leather and textile Moulded rubber sole Padded boot leg and insole Toe protection 6-hole lacing **Indoor athletic** Mass: 600 g Cow leather Texon Baking insole Moulded rubber sole (fine) Textile lining 6-hole lacing	Significant increase in stride length in combat boot compared to barefoot (*p* < 0.001). Greater increase in stride length in combat boot compared to outdoor shoe (*p* = 0.005). Significant reduction in plantar flexion in combat boot compared to barefoot (*p* < 0.001) and all other shoe types (*p* < 0.05). No significant change in knee ROM in combat boot compared to barefoot. No significant change in hip ROM in combat book compared to barefoot.	
Scott et al. [50] 2015	Cross Sectionsn = 195 Army Cadets (165 male, 30 female) Age range = 18 to 33 yr BMI = 23.5 ± 2.9 kg/m^2^	Most frequent boot type worn was collected through survey.		41 cadets suffered a lower extremity injury. 7 wearing government issued footwear. 17 wearing conventional running shoes. 17 wearing “other”. Boot type was not significantly associated with injury (χ^2^ = 0.19, *p* = 0.91).
Simeonov et al. [45] 2018	Quasi-experimental n = 24 male construction workers Age (Range) = 39 (23–53) yr Height = 178.3 ± 6.9 cm Mass = 86.4 ± 12.6 kg	6 shoe styles compared: **Running Shoe** Mass: 312 g Impact-peak acceleration (PG): 8.43 gravity unit for acceleration (G) Impact-time to peak acceleration (TTP): 14.20 ms Initial flex stiffness: 0.17 Nm/deg Torsion: 3.80 Nm Heel width: 8.5 cm **Tennis Shoe** Mass: 416 g Impact-PG: 8.59 G Impact-TTP: 12.60 ms Initial flex stiffness: 0.35 Nm/deg Torsion: 5.34 Nm Heel width: 9.1 cm **Basketball Shoe** Mass: 472 g Impact-PG: 8.79 G Impact-TTP: 11.20 ms Initial flex stiffness: 0.35 Nm/deg Torsion: 5.70 Nm Heel width: 9.2 cm **Work low-cut** Mass: 642 g Impact-PG: 8.76 G Impact-TTP: 10.10 ms Initial flex stiffness: 0.42 Nm/deg Torsion: 7.22 Nm Heel width: 8.7 cm **Work boot** Mass: 690 g Impact-PG: 10.05 G Impact-TTP: 9.27 ms Initial flex stiffness: 0.47 Nm/deg Torsion: 7.16 Nm Heel width: 8.7 cm **Safety boot** Mass: 768 g Impact-PG: 11.91 Impact-TTP: 9.43 ms Initial flex stiffness: 0.54 Nm/deg Torsion: 7.59 Nm Heel width: 8.7 cm	Significant main effects for footwear F_65, 490.7_ = 3.39 *p* < 0.0001 and the interaction of footwear and environment F_195, 3319.2_ = 1.66, *p* < 0.0001) Trunk angular velocity (T-AV): Similar while walking at simulated ground level High cut shoes resulted in significantly (*p* < 0.05) lower T-AV compared to low cut shoes on 15 and 25 cm planks Safety boots had the lowest T-AV on 15 cm plank Generally lower T-AV for work and safety boots compared to work shoes, but not significant T-AV less affected by plank width when wearing safety (*p* = 0.0516) or work (*p* = 0.0005) compared to low-cut work shoe Rearfoot angular velocity (F-AV): Similar F-AV between work shoes when walking on ground level High-cut shoes resulted in significantly (*p* < 0.05) less F-AV than low-cut shoes on 15 cm, but not 25 cm planks Surface lateral slope caused less increase in F-AV for work low-cut shoes compared to safety boots	Perceptions of instability (PI) Similar PI regardless of shoe when walking on simulated ground level Workers felt significantly more stable in high cut shoes (*p* < 0.05) compared to low cut shoes on 15 and 25 cm planks Work boots had a higher, but not significantly so, perception of comfort compared to other work shoes.
Sousa et al. [70] 2016	Case-Control n = 30 female hairdressers, 14 experiment and 16 in control **Experimental Group** Age = 34.6 ± 7.7 yr Mass = 65.3 ± 9.6 kg Height = 159.0 ± 6.0 cm **Control Group** Age = 34.9 ± 8.0 yr Mass = 61.1 ± 6.3 kg Height = 162.0 ± 6.0 cm	Unstable shoe with rounded sole use for 8 weeks compared to regular footwear.		Wearing unstable shoe for 8 weeks presented: Decreased in Centre of pressure area (F(1,27) = 8.296, *p* = 0.01) Decreased Medial-lateral centre of pressure root mean square (RMS) (F(1,27) =4.376, *p* = 0.046) Decreased Anteroposterior movement to a reference point (RM) peak to peak amplitude (F(1,27) = 8.414, *p* = 0.007) Increased Anteroposterior RM mean velocity (F(1,27) = 4.641, *p* = 0.040) Increased medial-lateral RM peak to peak amplitude (F(1,27) = 17.457, *p* < 0.001) Decreased anteroposterior RMS of body movement around a reference point (F(1,27) = 8.069, *p* < 0.001) Higher anteroposterior centre of pressure mean velocity (F(1,27) = 6.684, *p* = 0.015) Higher anteroposterior centre of pressure RMS (F(1,27) = 37.694, *p* < 0.001) Higher medial-lateral centre of pressure (F(1,27)—83.820, *p* < 0.001) RMS and area (F(1,27)—40.175, *p* < 0.001) Lower RM peak to peak amplitude anteroposterior (F(1,27) = 5.073, *p* < 0.001) and lower RMS anteroposterior F(1,27) = 21.667, *p* < 0.001) Higher RM peak to peak amplitude and higher RMS in medial-lateral (F(1,27) = 137.664, *p* < 0.001, F(1,27) = 11.084, *p* = 0.003) Higher medial lateral RM area (F(1,27) = 102.5334, *p* < 0.001) Reference through which the body oscillates (TR) was observed for anteroposterior RMS (F(1,27) = 18.704, *p* < 0.001) Increased medial-lateral TR RMS (F(1,27)—6.804, *p* = 0.015) Increased medial-lateral TR area (F(1,27) = 37.721, *p* < 0.001) Increased thigh antagonist co-activation (F(1,27) = 6.414, *p* = 0.012) Decreased thigh co-activation (F(1,28) = 21.038, *p* < 0.001) Increased reciprocal activation (F(1,28) = 18.23, *p* < 0.001) Decreased leg co-activation (F(1,28)—8.131, *p* = 0.008) Increased reciprocal activation (F(1,28) = 22.292, *p* < 0.001) Decreased lobal antagonist co-activation (F(1,28) = 12.940, *p* = 0.001) Increased total agonist activity (F(1,28) = 25.711, *p* < 0.001)
Svenningsen et al. [46] 2017	Quasi-experimental n = 14 working adults across various professions (7 male, 7 female) Age = 39.3 ± 6.8 yr Mass = 75.9 ± 12.6 kg Height = 175.7 ± 7.3 cm	**Unstable shoe** Rocker sole shoes Sole 2.0 cm at heel, 3.9 cm at midfoot, 3.5 cm in forefoot **Control** 1.9 cm height at heel, 3.0 cm in midfoot, 3.0 cm in forefoot	No main effects of shoes or load, or interaction effects between the two were found on stride duration or stride frequency.	Significantly higher EMG peak in longissimus thoracis wearing unstable shoes (*p* = 0.01, ηp^2^ = 0.143). Significantly higher RMS for longissimus thoracis (*p* = 0.001, ηp^2^ = 0.614) and iliocostalis lumborum (*p* = 0.0001, ηp^2^ = 0.487).
Talley et al. [61] 2009	Cohort n = 38 ship injury reports	Probability of ship injury wearing steel-toed safety boot versus not wearing such boots.		42.1% of individuals injured were wearing safety boots. Wearing safety boots was significantly likely to decrease probability of injury on container ship 2 (decrease by 0.609) (separated by container ship 1 by having different union officers).
Tojo et al. [51] 2018	Cross-sectional n = 636 nurses Mean demographic data not provided			Low shoe comfort score was associated with: Reported presence of foot and ankle pain (OR 2.12 (95% CI 1.31–3.50, *p* = 0.002) Presence of foot pain assessed with Manchester Foot Pain and Disability Index (OR 1.78 (95% CI 1.18–2.69) *p* = 0.006) Presence of disabling foot pain as assessed with Manchester Foot Pain and Disability Index (OR 1.76 (95% CI 1.01–3.11), *p* = 0.04) Medium comfort not significantly associated with presence of foot and ankle pain (*p* = 0.21).
Viera et al. [58] 2016	Randomised Control Trial n = 10 female nurses and 10 matched pairs **Control Group** Age = 31.0 ± 5.0 yr Mass = 66.0 ± 9.0 kg Height = 161.0 ± 5.0 cm **Experimental Group** Age = 34.0 ± 6.0 yr Mass = 68.0 ± 11.0 kg Height = 165.0 ± 7.0 cm	Unstable shoes compared to regular occupational footwear.		Experimental group reported significantly lower levels of pain at Weeks 4 (*p* = 0.016) and 6 (*p* < 0.001). Significantly lower levels of disability at Week 6 (*p* = 0.020). Participants started at moderate levels of disability at baseline and were at minimal post intervention.
Vu et al. [7] 2017	Quasi-experimental n = 20 male firefighters Age = 41.3 ± 8.8 yr Mass = 84.4 ± 11.6 kg Height = 181.0 ± 6.0 cm	Comparison between firefighting boots and athletic footwear. **Firefighting boots** 178 mm shaft height in EU size 41 or 42 192 mm shaft size for size 45 and above Type 2 structural Haix Fire Flash Xtreme boot		Landing in firefighter boots resulted in: Significantly increased vertical GRF (2.40 times BW compared to 2.14 times, *p* < 0.05) Reduction of right ankle plantarflexion angle in unloaded (control −36.41 ± 10.43°, boot −25.59 ± 9.17°) and loaded comparisons (control −40.73 ± 11.55°, boot −28.31± 13.51°) *p* < 0.05 Greater peak lumbopelvic flexion angular velocity while unloaded (222.89 ± 18.47 °/s) and load (240.91 ± 24.37 °/s) compared to control unloaded (187.56 ± 60.96 °/s), *p* < 0.05 Increased peak lumbopelvic flexion force (unloaded 13.40 ± 1.48 N/kg, loaded 14.46 ± 1.46 N/kg) compared to control (unloaded 12.32 ± 5.44 N/kg, loaded 14.37 ± 6.94 N/kg), *p* < 0.05 Greater peak lumbopelvic adduction force (control 19.84 ± 11.44 N/kg, boot 30.54 ± 19.76 N/kg), *p* < 0.01 Greater peak lumbopelvic adduction moments (control 2.21 ± 1.01 Nm/kg, boots 3.15 ± 0.50 Nm/kg), *p* < 0.001 Greater peak lumbar internal rotation angular velocities unloaded (111.06 ± 13.07 °/s) and loaded (114.49 ± 16.40 °/s) compared to unloaded control (80.51 ± 47.23 °/s), *p* < 0.05 Greater peak lumbar internal rotation moments unloaded (3.26 ± 0.34 Nm/kg) and loaded (3.81 ± 0.42 Nm/kg) compared to control (2.60 ± 0.78 Nm/kg) *p* < 0.05
Werner et al. [52] 2010	Cross sectional study n = 407 automotive workers Age = 48.4 ± 10.3 yr BMI = 29.4 ± 5.3 kg/m^2^			Rotation of shoes during the work week reduced risk of presenting with plantar fasciitis (OR 0.30, *p* = 0.01, 95% CI 0.1–0.7). No significant effect of outer sole stiffness on plantar fasciitis.
Werner et al. [54] 2010	Cross-sectional study n = 407 automotive workers Age = 48.4 ± 10.3 yr BMI = 29.4 ± 5.3 kg/m^2^			Shoe rotation not significantly associated with foot/ankle disorders (*p* = 0.75). Outer sole stiffness not significantly associated with foot/ankle disorders (*p* = 0.77) but was associated with new foot and ankle disorders: Stiffness middle tertile OR 8.2, *p* = 0.05, 95% CI 1.01–65.5 Stiffness upper tertile OR 18.9, *p* = 0.01, 95% CI 2.2–165.8
Werner et al. [53] 2011	Cross-sectional study n = 407 automotive workers Age = 48.4 ± 10.3 years BMI = 29.4 ± 5.3 kg/m^2^			No significant association between firmness of heel (*p* = 0.75), firmness of insole (*p* = 0.91) or shoe rotation (*p* = 0.35) and incidence of hip disorders.
Yeung et al. [66] 2011	Systematic Review Military personnel	Identified two studies that compared a tropical combat boot cotton/nylon blend to a leather combat boot.		No significant difference between footwear and lower limb soft-tissue injuries for any location.

All descriptive values represent mean and standard deviation unless otherwise noted. ANOVA: Analysis of Variance; ANCOVA: Analysis of Covariance; BMI: Body Mass Index; BW: Body weight; CAMPD: Cumulative muscle activity per distance travelled; CI: Confidence Interval; EMG: Electromyography; EVA: Ethylene-vinyl acetate; MIN: Minimalist; NFPA: National Fire Protection Association; OR: Odds Ratio; PPE: Personal Protective Equipment; PU: Polyurethane; RCT: Randomised Control Trial; RER: Respiratory Exchange Ratio; RMS: Root Mean Square; ROM: Range Of Movement; RPE: Rating Of Perceived Exertion; RR: Risk Ratio; SBR: Styrene-butadiene rubber; VO_2_: Volume of Oxygen; VCO_2_: Volume of Carbon Dioxide; VE: Ventilatory Exchange, Yr: year.

## Data Availability

Not applicable.

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
