# Peer review of "The Impact of Footwear on Occupational Task Performance and Musculoskeletal Injury Risk: A Scoping Review to Inform Tactical Footwear"

_ijerph, 2022, doi:10.3390/ijerph191710703_

Round 1
Reviewer 1 Report
General Comments
This manuscript provides a scoping review of the impact of footwear on occupational performance and injury risk outcomes. The manuscript is well written and systematically reviews relevant literature. The need for a review of this topic is clearly described as this manuscript will inform future research and ultimately the development of appropriate footwear to enhance occupational safety and performance.
Specific Comments
Abstract:
L2: The use of the term “Occupational” seems redundant in the title. Consider omitting first “Occupational” term.
L9: Revise “impacts” to “impact”.
L18 & 19: Remove the comma after “etc.”.
L20: Remove “on” to enhance readability.
L21 & 22: Replace “like” with “e.g.,”.
L22: Consider omitting parenthetical content and revise to: …“and select occupational tasks.”.
Introduction
L33: Revise to “law enforcement officers”.
L56: Revise to: “For example, sole hardness in footwear can impact”…
L58-59: Revise to: … “that the boots are designed to mitigate injuries, not to become a cause of them.”
L60-62: Rephrase sentence for readability.
L65: Revise to: “Further, the impact of”…
Materials and Methods
Table 1
Add a period after all table figure titles.
Table 2
Revise first bullet point under inclusion criteria to: military “personnel”.
L105: Studies investigating slips, trips, falls were excluded. However, Chiou et al. (2012) was included in Table 3.
L105: Please clarify what subjective report refers to. L373 describes study results that utilized subjective feedback on footwear (Ref #45). Should this study be included?
Also, some of the studies cited in the narrative / systematic reviews were published outside of 20 years. Please consider this as it may / may not relate to the inclusion / exclusion criteria. Perhaps only the publication date of the review article qualifies for inclusion.
5th bullet under “Exclusion” criteria: Revise “risks” to “risk”.
L108: Describe if the searching, screening and selection process was conducted manually? If so, how many investigators participated in the process? Additionally, was inter-observer reliability for study inclusion and data extraction evaluated?
Results
L126-127: Should conference abstracts be used in a scoping review? It would seem that the peer-review process may not be as rigorous compared to the peer-review process for a manuscript.
Figure 1: Screening section: Indicates that articles not published within 15 years were excluded. L96 indicates that publications in the last 20 years were included. Please clarify and revise accordingly.
L144: Should studies be included that used “healthy adults” instead of an occupational population?
L157-158: Revise for readability to: … “and their impact on gait mechanics, joint range of motion,”…
L164: Consider rephrasing heading to: “Impact of Footwear on Occupational Performance”.
L176, L222, : Here and throughout the manuscript “impacts” is used where “impact” seems appropriate.
L183: Be consistent with heading phrases. Consider: “Impact of Footwear on Spatiotemporal and Angular Velocities of Gait”. -Capitalize “Gait”.
L243-245: Please provide a brief interpretation of this statement for the reader.
L282: Revise to: … “Oschman et al. [9] compared the effects of various occupational footwear on plantar pressure in automotive workers.”
L297: Consider rephrasing to: “Impact of Occupational Footwear on Joint Range of Motion”.
L301: Revise to: … “military personnel”…
L303-304: Revise “weight” to “mass” here and as appropriate throughout the manuscript. Most values provided reflect the mass of the footwear.
L346-347, 361: Add appropriate symbols to reflect p-values.
L366: Consider rephrasing the heading. Consider: “Impact of Occupational Footwear on Posture and Balance”.
L374: Revise to: …“perceptions of stability”…
L394-396: Please describe what the “±” represents. Presumably these are mean footwear mass values, however it reads as standard deviation values.
L401: Consider: “Impact of Occupational Footwear on Physiological Outcomes”.
L411: Revise “5-min” to “5 min” for consistency of formatting values and units throughout the manuscript.
L429: Revise “great” to “greater”.
L436: Revise “30 minutes” to “30 min” for formatting consistency.
L444: “distance of the foot from the centred mass of the body” sounds odd. Consider rephrasing or clarify this measurement.
L500-501: Remove generic muscle names. Utilize anatomical muscle names for this audience.
L537: Revise to: “Impact of Occupational Footwear on Musculoskeletal Injury Risk”.
L545-547: This content reads very similar to L538-542, although different references are cited. Please clarify difference or omit content for the reader.
L598: Revise to: … “with the wearing of safety boots serving as a predictive factor.”
L613: Here and throughout the manuscript, if a study reported a nonsignificant trend of change between footwear conditions, it would be helpful to provide the reader with the p-value and/or effect size to interpret the trend. -see also Armand et al. [55] in Table 3 under “Impact on Injury Risk” column.
Table 3: Especially given the extensive length of this article, there appears to be substantial redundancy between the content in text and in this table. Is this necessary?
-Insert a period after table title.
-To enhance readability, consider noting under the table that all values represent mean and SD, unless noted.
-Insert “:” after descriptive variables in the Study Design column (i.e., Age: 29.3…). Round all values to 1 decimal place.
-Revise “weight” to “mass”. Revise “yrs” to “yr”. Place punctuation (i.e., “,”) after each variable in Study Design column.
-Delete the parenthesis symbol after “leather shoes” in Boot Construction column in Al-Ashaik et al. [27] study. Also, revise “shoot” to “boot” under the Impact on Task Performance colunn of this study.
-Under Chander et al. [63] revise “lower heel heigh” to “height”.
-Under Chiou et al. [30], “Impact on Task Performance” column add “)” to Greater VE bullet point. Revise to: “Greater absolute VO2” on the bullet point immediately below.
-Choukou et al. [68-69]: Should conference abstracts be included in the analysis?
-Choukou et al. [69]: Use lower case letters for variables under Impact on Task Performance. Correct spelling of “Chukou” here.
-Dobson et al. [13]: Delete space under Impact on Task Performance after “boots ,”.
-Dobson et al. [59]: Revise “minders” to “miners”.
-Dobson et al. [47]: Under “Impact on Injury Risk”, revise to “No significant difference between boots for”.
-Dobson et al. [31]: Under “Study Design” column, revise “trades workers” to “trade workers”. Place “:” after all variable names (i.e., Mass: 0.94 kg). Do so throughout Table 3.
-Elbers et al. [57]: Revise “seconds” to “sec” or “s” as unit abbreviations have been used throughout the manuscript.
-Gell et al. [49]: Revise “didn’t” to “did not”.
-Huebener et al. [33]: Under Impact on Task Performance column, revise to … “and fast walking velocities”. Add punctuation after all bullet points (i.e., ;) here and throughout table.
Discussion
L685-701: To enhance readability, omit the parenthetical information from this paragraph. It reads like the abstract. I appreciate the authors’ effort to add clarity, however, specific details can be provided in the Discussion below.
L685-687: Consider rephasing to: …“literature regarding the impact of occupational footwear on task performance...
L693: Remove "on". Were all of these outcomes "negative". If so, state as such.
L709: Revise “phase” to “phases”.
L710-714: Run-on sentence. Revise for readability.
L718: Revise “weighted” to “weighed”.
L737: Revise to “Greater boot shaft height and stiffness…”.
L757: Revise “range” to “ROM”.
L886-894: Did Al-Ashaik et al. [27] speculate about the mechanism by which there is an interaction between temperature and shoe mass on upper body muscle activation?
L898-906: Interesting. How was the presumable variability in ambulatory speed controlled for across conditions?
L929: Revise to … “types of boots worn did not impact injury risk.”
L938-940: Omit sentence to enhance readability.
L944-945: Revise “risks” to “risk”.
L945: Revise “rate” to “rated”.
L965: Spell out heart rate.
L980: Revise to: … “investigated the impact of footwear…”.
L997-998: I believe polyurethane and/or styrene-butadiene were previously abbreviated terms.
L1000: Revise “impacts” to “impact”.
Author Response
We would like to thank Reviewer 1 for their time and dedicated efforts to improve this manuscript. The effort and diligence of their review is appreciated. Please see our responses to this effort below.
General Comments
This manuscript provides a scoping review of the impact of footwear on occupational performance and injury risk outcomes. The manuscript is well written and systematically reviews relevant literature. The need for a review of this topic is clearly described as this manuscript will inform future research and ultimately the development of appropriate footwear to enhance occupational safety and performance.
Specific Comments
Abstract:
L2: The use of the term “Occupational” seems redundant in the title. Consider omitting first “Occupational” term.
- Thank you. Amended as suggested.
L9: Revise “impacts” to “impact”.
- Thank you. Amended as suggested.
L18 & 19: Remove the comma after “etc.”.
- Thank you. Amended as suggested.
L20: Remove “on” to enhance readability.
- Thank you. Amended as suggested.
L21 & 22: Replace “like” with “e.g.,”.
- Thank you. Amended as suggested.
L22: Consider omitting parenthetical content and revise to: …“and select occupational tasks.”.
- Thank you. Amended as suggested.
Introduction
L33: Revise to “law enforcement officers”.
- Thank you. Amended as suggested.
L56: Revise to: “For example, sole hardness in footwear can impact”…
- Thank you. Amended as suggested.
L58-59: Revise to: … “that the boots are designed to mitigate injuries, not to become a cause of them.”
- Thank you. Amended as suggested.
L60-62: Rephrase sentence for readability.
- Thank you. Amended as suggested.
L65: Revise to: “Further, the impact of”…
- Thank you. Amended as suggested.
Materials and Methods
- Thank you. Amended as suggested.
Table 1
Add a period after all table figure titles.
- Thank you. Amended as suggested.
Table 2
Revise first bullet point under inclusion criteria to: military “personnel”.
- Thank you. Amended as suggested.
L105: Studies investigating slips, trips, falls were excluded. However, Chiou et al. (2012) was included in Table 3.
- Thank you. This has been clarified. Studies investigating slips, trips, and falls in this context were those that focussed specifically on the mechanism of injury and may have mentioned footwear as a causative (secondary factor) as opposed to investigating footwear and their relationships with slips, trips, and falls (as did Chiou et al. (2012)) where footwear is a primary factor informing the research. Wording in text has been clarified to: ‘The study primarily investigated risk of slips, trips, and falls with footwear potentially discussed as a causative factor;’
L105: Please clarify what subjective report refers to. L373 describes study results that utilized subjective feedback on footwear (Ref #45). Should this study be included?
- Thank you. Studies reporting only subjective findings were excluded. The included study by Simeonov et al. 2018, included objective measures. Wording in the exclusion has been amended to: ‘The study only reported on subjective findings / worker feedback on footwear’.
Also, some of the studies cited in the narrative / systematic reviews were published outside of 20 years. Please consider this as it may / may not relate to the inclusion / exclusion criteria. Perhaps only the publication date of the review article qualifies for inclusion.
- The oldest included article is ‘Majumdar D, Banerjee PK, Majumdar D, Pal M, Kumar R, Selvamurthy W. Temporal spatial parameters of gait with barefoot, bathroom slippers and military boots. Indian J Physiol Pharmacol 50(1):33-40, 2006.’ Which was 15 years old when the search weas done in OCT 2021.
5th bullet under “Exclusion” criteria: Revise “risks” to “risk”.
- Thank you. Amended as suggested.
L108: Describe if the searching, screening and selection process was conducted manually? If so, how many investigators participated in the process? Additionally, was inter-observer reliability for study inclusion and data extraction evaluated?
- Thank you this has now been added.
Results
L126-127: Should conference abstracts be used in a scoping review? It would seem that the peer-review process may not be as rigorous compared to the peer-review process for a manuscript.
- Thank you. For scoping reviews which serves as a knowledge synthesis, follow a systematic approach to map evidence on a topic and identify main concepts, theories, sources, and knowledge gaps’ (Tricco, Andrea C., Erin Lillie, Wasifa Zarin, Kelly K. O'Brien, Heather Colquhoun, Danielle Levac, David Moher et al. "PRISMA extension for scoping reviews (PRISMA-ScR): checklist and explanation." Annals of internal medicine 169, no. 7 (2018): 467-473.), scoping reviews can include any form of literature – even media articles.
Figure 1: Screening section: Indicates that articles not published within 15 years were excluded. L96 indicates that publications in the last 20 years were included. Please clarify and revise accordingly.
- Thank you. The oldest study we identified was 15 years old – the error is in the PRISMA and has been corrected.
L144: Should studies be included that used “healthy adults” instead of an occupational population?
- Thank you. Yes, these studies had healthy adults who were working doing occupational tasks (e.g., long standing, walking, climbing stairs etc.)
L157-158: Revise for readability to: … “and their impact on gait mechanics, joint range of motion,”…
- Thank you. Amended as suggested.
L164: Consider rephrasing heading to: “Impact of Footwear on Occupational Performance”.
- Thank you. Amended as suggested.
L176, L222, : Here and throughout the manuscript “impacts” is used where “impact” seems appropriate.
- Thank you. These has been reviewed and ‘impacts’ retained where more than one are being discussed and singular ‘impact’ as appropriate.
L183: Be consistent with heading phrases. Consider: “Impact of Footwear on Spatiotemporal and Angular Velocities of Gait”. -Capitalize “Gait”.
- Thank you. Corrected throughout
L243-245: Please provide a brief interpretation of this statement for the reader.
- Thank you. Interpretation added. ‘There was also a significantly higher (p < 0.01) median power frequency in the PU boots (24.6±2.8 Hertz [Hz] unloaded, 24.9±2.2 Hz loaded) compared to SBR boots (21.8±3.3 Hz unloaded, 22.4±4.7 Hz loaded) [38], suggesting higher forces throughout the duration of the measured stride.’
L282: Revise to: … “Oschman et al. [9] compared the effects of various occupational footwear on plantar pressure in automotive workers.”
- Thank you. Amended as suggested.
L297: Consider rephrasing to: “Impact of Occupational Footwear on Joint Range of Motion”.
- Thank you. Amended as suggested.
L301: Revise to: … “military personnel”…
- Thank you. Amended as suggested.
L303-304: Revise “weight” to “mass” here and as appropriate throughout the manuscript. Most values provided reflect the mass of the footwear.
- Thank you. We have made changes as requested. However, technically, these are all weights as they are measured using scales and as such are relative to gravity not composition.
L346-347, 361: Add appropriate symbols to reflect p-values.
- Thank you. Amended as suggested.
L366: Consider rephrasing the heading. Consider: “Impact of Occupational Footwear on Posture and Balance”.
- Thank you. Amended as suggested.
L374: Revise to: …“perceptions of stability”…
- Thank you. Amended as suggested.
L394-396: Please describe what the “±” represents. Presumably these are mean footwear mass values, however it reads as standard deviation values.
- Thank you. Corrected to ‘~’
L401: Consider: “Impact of Occupational Footwear on Physiological Outcomes”.
- Thank you. Amended as suggested.
L411: Revise “5-min” to “5 min” for consistency of formatting values and units throughout the manuscript.
- Thank you. Amended as suggested.
L429: Revise “great” to “greater”.
- Thank you. Amended as suggested.
L436: Revise “30 minutes” to “30 min” for formatting consistency.
- Thank you. Amended as suggested.
L444: “distance of the foot from the centred mass of the body” sounds odd. Consider rephrasing or clarify this measurement.
- Thank you. Reworded: ‘Thus, the authors concluded that wearing boots led to greater heat production as the distance of the foot from the body’s centre of mass makes heat distribution mechanically inefficient.’
L500-501: Remove generic muscle names. Utilize anatomical muscle names for this audience.
- Thank you. Amended as requested.
L537: Revise to: “Impact of Occupational Footwear on Musculoskeletal Injury Risk”.
- Thank you. Amended as suggested.
L545-547: This content reads very similar to L538-542, although different references are cited. Please clarify difference or omit content for the reader.
- Thank you. This has been clarified as the numbering system for the two following sections was incorrect. The content in L538 to 542 segues into 3.11.1 and 3.11.2
L598: Revise to: … “with the wearing of safety boots serving as a predictive factor.”
- Thank you. Amended as suggested.
L613: Here and throughout the manuscript, if a study reported a nonsignificant trend of change between footwear conditions, it would be helpful to provide the reader with the p-value and/or effect size to interpret the trend. -see also Armand et al. [55] in Table 3 under “Impact on Injury Risk” column.
- Thank you. We have only presented non-significant p values where they clarify information. If all non-significant values were added the table size would increase exponentially.
Table 3: Especially given the extensive length of this article, there appears to be substantial redundancy between the content in text and in this table. Is this necessary?
- Thank you. We are trying to walk the fine line between providing sufficient information in the text to allow understanding without having to constantly refer to the table while also appreciating that for some the table is a useful single point summation of the work.
-Insert a period after table title.
- Thank you. Amended as suggested.
-To enhance readability, consider noting under the table that all values represent mean and SD, unless noted.
- Thank you. Amended as suggested.
-Insert “:” after descriptive variables in the Study Design column (i.e., Age: 29.3…). Round all values to 1 decimal place.
- Thank you. Amended as suggested.
-Revise “weight” to “mass”. Revise “yrs” to “yr”. Place punctuation (i.e., “,”) after each variable in Study Design column.
- Thank you. Amended as suggested.
-Delete the parenthesis symbol after “leather shoes” in Boot Construction column in Al-Ashaik et al. [27] study. Also, revise “shoot” to “boot” under the Impact on Task Performance colunn of this study.
- Thank you. Amended as suggested.
-Under Chander et al. [63] revise “lower heel heigh” to “height”.
- Thank you. Amended as suggested.
-Under Chiou et al. [30], “Impact on Task Performance” column add “)” to Greater VE bullet point. Revise to: “Greater absolute VO2” on the bullet point immediately below.
- Thank you. Amended as suggested.
-Choukou et al. [68-69]: Should conference abstracts be included in the analysis?
- As clarified earlier in responses.
-Choukou et al. [69]: Use lower case letters for variables under Impact on Task Performance. Correct spelling of “Chukou” here.
- Thank you. Amended as suggested.
-Dobson et al. [13]: Delete space under Impact on Task Performance after “boots ,”.
- Thank you. Amended as suggested.
-Dobson et al. [59]: Revise “minders” to “miners”.
- Thank you. Amended as suggested.
-Dobson et al. [47]: Under “Impact on Injury Risk”, revise to “No significant difference between boots for”.
- Thank you. Amended as suggested.
-Dobson et al. [31]: Under “Study Design” column, revise “trades workers” to “trade workers”. Place “:” after all variable names (i.e., Mass: 0.94 kg). Do so throughout Table 3.
- Thank you. Corrected.
-Elbers et al. [57]: Revise “seconds” to “sec” or “s” as unit abbreviations have been used throughout the manuscript.
- Thank you. Amended as suggested.
-Gell et al. [49]: Revise “didn’t” to “did not”.
- Thank you. Amended as suggested.
-Huebener et al. [33]: Under Impact on Task Performance column, revise to … “and fast walking velocities”. Add punctuation after all bullet points (i.e., ;) here and throughout table.
- Thank you. Corrected. However, we have not added punctuation to the end of bullet points as per academic writing conventions which differ from business writing conventions, (see Office for National Statistics). We have however gone back and ensured punctuation consistency throughout the table.
Discussion
L685-701: To enhance readability, omit the parenthetical information from this paragraph. It reads like the abstract. I appreciate the authors’ effort to add clarity, however, specific details can be provided in the Discussion below.
- Thank you. Amended as suggested.
L685-687: Consider rephasing to: …“literature regarding the impact of occupational footwear on task performance...
- Thank you. Amended as suggested.
L693: Remove "on". Were all of these outcomes "negative". If so, state as such.
- Amended as suggested. No, not all outcomes were negative (e.g., changes to gait were changes that were neither positive nor negative, they just differed when compared to a baseline).
L709: Revise “phase” to “phases”.
- Thank you. Amended as suggested.
L710-714: Run-on sentence. Revise for readability.
- Thank you. Amended as suggested.
L718: Revise “weighted” to “weighed”.
- Thank you. Amended as suggested.
L737: Revise to “Greater boot shaft height and stiffness…”.
- Thank you. Amended as suggested.
L757: Revise “range” to “ROM”.
- Thank you. Amended as suggested.
L886-894: Did Al-Ashaik et al. [27] speculate about the mechanism by which there is an interaction between temperature and shoe mass on upper body muscle activation?
- Indirectly – Upper body muscle activity was related to the amount of work that could be performed (number of lifts per unit of time) which was in turn influenced by the shoes and the temperature.
L898-906: Interesting. How was the presumable variability in ambulatory speed controlled for across conditions?
- By a timed pace setter using 400m checkpoints to confirm speed.
L929: Revise to … “types of boots worn did not impact injury risk.”
- Thank you. Amended as suggested.
L938-940: Omit sentence to enhance readability.
- Thank you. Amended as suggested.
L944-945: Revise “risks” to “risk”.
- Thank you. Amended as suggested.
L945: Revise “rate” to “rated”.
- Thank you. Amended as suggested.
L965: Spell out heart rate.
- Thank you. Amended as suggested.
L980: Revise to: … “investigated the impact of footwear…”.
- Thank you. Amended as suggested.
L997-998: I believe polyurethane and/or styrene-butadiene were previously abbreviated terms.
- Thank you. Amended as suggested.
L1000: Revise “impacts” to “impact”.
- Thank you. Amended as suggested.
Reviewer 2 Report
Title: The Impact of Occupational Footwear on Occupational Task Performance and Musculoskeletal Injury Risk: A Scoping Review to Inform Tactical Footwear.
This article seems well built and brings evidence of a physiological phenomenon not yet fully understood and that certainly deserves further study.
Some hard points of revision, suggested to the authors to increase the level of the paper, are provided below
Unclear, if this manuscript is a Scoping Review or Systematic Review, because in some part of this submission the authors Used the term “Systematic”
I suggest following the SYSTEMATIC REVIEW AND META‐ANALYSIS: A PRIMER by Impellizzeri & Bizzini https://www.ncbi.nlm.nih.gov/pmc/articles/PMC3474302/
In fact, their suggest some details for META‐ANALYSIS:
- Data extraction must be accurate and unbiased and therefore, to reduce possible errors, it should be performed by at least two researchers. Standardized data extraction forms should be created, tested, and if necessary modified before implementation. The extraction forms should be designed taking into consideration the research question and the planned analyses. Information extracted can include general information (author, title, type of publication, country of origin, etc.), study characteristics (e.g. aims of the study, design, randomization techniques, etc.), participant characteristics (e.g. age, gender, etc.), intervention and setting, outcome data and results (e.g. statistical techniques, measurement tool, number of follow up, number of participants enrolled, allocated, and included in the analysis, results of the study such as odds ratio, risk ratio, mean difference and confidence intervals, etc.). Disagreements should be noted and resolved by discussing and reaching a consensus. If needed, a third researcher can be involved to resolve the disagreement.
Author Response
We would like to thank Reviewer 2 for their time and consideration. Please see our responses to clarify their feedback below.
This article seems well built and brings evidence of a physiological phenomenon not yet fully understood and that certainly deserves further study.
Some hard points of revision, suggested to the authors to increase the level of the paper, are provided below
Unclear, if this manuscript is a Scoping Review or Systematic Review, because in some part of this submission the authors Used the term “Systematic”
I suggest following the SYSTEMATIC REVIEW AND META‐ANALYSIS: A PRIMER by Impellizzeri & Bizzini https://www.ncbi.nlm.nih.gov/pmc/articles/PMC3474302/
In fact, their suggest some details for META‐ANALYSIS:
- Data extraction must be accurate and unbiased and therefore, to reduce possible errors, it should be performed by at least two researchers. Standardized data extraction forms should be created, tested, and if necessary modified before implementation. The extraction forms should be designed taking into consideration the research question and the planned analyses. Information extracted can include general information (author, title, type of publication, country of origin, etc.), study characteristics (e.g. aims of the study, design, randomization techniques, etc.), participant characteristics (e.g. age, gender, etc.), intervention and setting, outcome data and results (e.g. statistical techniques, measurement tool, number of follow up, number of participants enrolled, allocated, and included in the analysis, results of the study such as odds ratio, risk ratio, mean difference and confidence intervals, etc.). Disagreements should be noted and resolved by discussing and reaching a consensus. If needed, a third researcher can be involved to resolve the disagreement.
- Thank you. Please note that this was a scoping review and followed the recommended Preferred Reporting Items for Systematic reviews and Meta-Analyses extension for Scoping Reviews (PRISMA-ScR) Checklist (https://prisma-statement.org/documents/PRISMA-ScR-Fillable-Checklist_11Sept2019.pdf). As noted in the checklist document referred to it is different to traditional systematic reviews and meta analysis being ‘…., a type of knowledge synthesis, follow a systematic approach to map evidence on a topic and identify main concepts, theories, sources, and knowledge gaps’ (Tricco, Andrea C., Erin Lillie, Wasifa Zarin, Kelly K. O'Brien, Heather Colquhoun, Danielle Levac, David Moher et al. "PRISMA extension for scoping reviews (PRISMA-ScR): checklist and explanation." Annals of internal medicine 169, no. 7 (2018): 467-473.). In essence while following a systematic approach it is not a systematic review – otherwise the standard PRISMA checklist (https://prisma-statement.org/documents/PRISMA_2020_checklist.pdf would have been followed.
- See also 11.1.1 Why a scoping review? - JBI Manual for Evidence Synthesis - JBI Global Wiki (refined.site) form the Joanna Briggs Institute… ‘Unlike other reviews that tend to address relatively precise questions (such as a systematic review of the effectiveness of an intervention assessed using a predefined set of outcomes), scoping reviews can be used to map the key concepts that underpin a field of research, as well as to clarify working definitions, and/or the conceptual boundaries of a topic.’
Examples of other similar scoping reviews include:
- Barbeau P, Michaud A, Hamel C, et al. Musculoskeletal Injuries Among Females in the Military: A Scoping Review. Military medicine. 2021;186(9-10):e903-e931. doi:10.1093/milmed/usaa555
- Kerr NC, Ashby S, Gerardi SM, Lane SJ. Occupational therapy for military personnel and military veterans experiencing post‐traumatic stress disorder: A scoping review. Australian occupational therapy journal. 2020;67(5):479-497. doi:10.1111/1440-1630.12684
We did however add some further clarification:
‘The screening process was conducted by one author (D.M.) and confirmed by a fellow author (R.P.) with consensus, where required, settled by a third author (R.O.).’ and ‘Data were extracted by one author (D.M.) and confirmed by fellow author (R.P.) with consensus, if required, settled by a third author (R.O.).’
Round 2
Reviewer 2 Report
Nothing